# BREAKING GRADIENT TEMPORAL COLLINEARITY FOR ROBUST SPIKING NEURAL NETWORKS

**Desong Zhang, Jia Hu,**\* and **Geyong Min**\*
Department of Computer Science, University of Exeter, Exeter, U.K.
`{dz288, J.Hu, G.Min}@exeter.ac.uk`

## ABSTRACT

Spiking Neural Networks (SNNs) have emerged as an efficient neuromorphic computing paradigm, offering low energy consumption and strong representational capacity through binary spike-based information processing. However, their performance is heavily shaped by the input encoding method. While direct encoding has gained traction for its efficiency and accuracy, it proves less robust than traditional rate encoding. To illuminate this issue, we introduce Gradient Temporal Collinearity (GTC), a principled measure that quantifies the directional alignment of gradient components across time steps, and we show—both empirically and theoretically—that elevated GTC in direct encoding undermines robustness. Guided by this insight, we propose **S**tructured **T**emporal **O**rthogonal **D**ecorrelation (STOD), which integrates parametric orthogonal kernels with structured constraints into the input layer of direct encoding to diversify temporal features and effectively reduce GTC. Extensive experiments on visual classification benchmarks, show that STOD consistently outperforms state-of-the-art methods in robustness, highlighting its potential to drive SNNs toward safer and more reliable deployment. Code available at `https://github.com/Apple26419/SNN_STOD`.

## 1 INTRODUCTION

Spiking Neural Networks (SNNs), as an emerging neuromorphic computing paradigm, process information over time through binary spikes (Maass, 1997), exhibiting outstanding efficiency and low power consumption in domains such as autonomous driving (Viale et al., 2021; Zhu et al., 2024), medical image processing (Pan et al., 2025; Liu et al., 2025), edge computing (Zhang et al., 2024; Liu et al., 2024a), and robot control (Kumar et al., 2025; Jiang et al., 2025). However, under binary spike-based paradigm, the performance of SNNs is largely determined by the input encoding method, as it directly shapes the network's temporal statistics and representational capacity (Auge et al., 2021).

In early studies, rate encoding was the most commonly adopted method (Heeger et al., 2000; Lee et al., 2016). Its core idea is to represent inputs by the firing frequency of randomly generated spikes within a temporal window, thereby transforming static data into spike trains with temporal attributes (Lee et al., 2020). However, rate encoding requires substantially long spike sequences to capture features with sufficient fidelity, and under the mainstream training paradigm of Backpropagation Through Time (BPTT), the increased sequence length dramatically amplifies computational cost, which severely limits efficiency in large-scale applications. To overcome this efficiency bottleneck, direct encoding has been proposed (Rueckauer et al., 2017). Unlike rate encoding, direct encoding injects identical original data into the network over only a few time steps, thereby maximally preserving the original features while significantly reducing the required sequence length. This property grants direct encoding distinct advantages in both training and inference performance, establishing it as the mainstream choice for efficient and high-performance SNN training (Fang et al., 2021).

In addition to efficiency and high performance, robustness during real-world deployment is equally critical for SNNs (Ding et al., 2024a;b; 2022; Kundu et al., 2021). Due to their cumulative membrane-potential dynamics, small input perturbations can be repeatedly propagated and amplified over time, making SNNs inherently more vulnerable to temporal perturbation accumulation than traditional artificial neural networks (Stanojevic et al., 2024). In many of the safety-critical domains where SNNs are most attractive, such as autonomous driving, robotics, and edge intelligence, this temporal sensitivity makes robustness not an optional enhancement but a fundamental requirement.

---

\*Corresponding author.

Despite being the mainstream choice for efficient and high-performance SNN training, direct encoding exhibits surprisingly worse robustness compared to the more traditional rate encoding (Kundu et al., 2021; Kim et al., 2022). Because identical inputs are repeatedly injected across time steps, membrane potentials accumulate highly correlated signals, causing the network to degenerate into an enlarged static feature extractor rather than exploiting temporal dynamics to capture complementary information. As a result, the lack of temporal diversity makes the network highly vulnerable to small input perturbations, which are repeatedly accumulated and amplified across time steps, rendering its representations fragile. In contrast, the randomized spike generation of rate encoding serves as a natural feature decorrelation mechanism: spike patterns across different time steps are independent, preventing perturbations from remaining consistent across all steps and thereby mitigating error accumulation while strengthening resistance to adversarial perturbations.

To formalize and gain deeper insight into the robustness differences between the two encoding methods, this paper approaches the problem from the perspective of training dynamics and introduces the concept of Gradient Temporal Collinearity (GTC). By quantifying the directional consistency of gradient components across time steps, GTC establishes a unified lens through which the robustness disparities of different encoding methods can be revealed and their underlying mechanisms rigorously explained. Building on GTC, we propose a novel robustness enhancement strategy, Structured Temporal Orthogonal Decorrelation (STOD), which incorporates a feature decorrelation mechanism inspired by rate encoding into direct encoding. By diversifying input features, STOD effectively reduces the level of GTC, thereby improving network robustness. Specifically, the contributions of this paper can be summarized as follows:

- **We introduce and quantify gradient temporal collinearity (GTC) in SNNs.** GTC provides a principled metric that reveals robustness disparities between direct and rate encoding and elucidates the mechanisms by which these disparities affect network robustness.

- **We propose Structured Temporal Orthogonal Decorrelation (STOD).** STOD introduces parametric orthogonal kernels with structured constraints into the input layer, which structurally breaks GTC and enhances the network's ability to resist external perturbations.

- Through extensive experiments on datasets of varying scales, we show that STOD consistently improves the robustness of SNNs without incurring extra inference overhead, and achieves superior robustness performance compared with existing state-of-the-art methods.

## 2 RELATED WORK

**Robustness differences across encoding methods.** Rate encoding is widely observed to provide higher robustness compared with direct encoding. Kundu et al. (2021) explained this advantage through proposed spike-based sparse activation maps, Mukhoty et al. (2025) associated it with the injection of noise through rate encoding, while Kim et al. (2022) experimentally demonstrated that rate encoding suffers less performance degradation under attacks. Thus, enhancing the robustness of SNNs under direct encoding remains a pressing challenge of both theoretical and practical importance.

**Robustness enhancement under direct encoding.** Within the more efficient direct encoding paradigm, several methods have been proposed to strengthen robustness: Ding et al. (2022) introduced a regularized adversarial training scheme to constrain the spiking Lipschitz constant of SNNs and thereby mitigate the impact of perturbations on network outputs; Ding et al. (2024a) proposed modifying LIF neurons and minimizing the mean squared perturbation of membrane potentials to stabilize SNNs; Geng & Li (2023) developed LIF neurons with adaptive thresholds that anchor membrane potentials to a neural dynamic signature, thereby minimizing perturbation errors; Liu et al. (2024b) enhanced adversarial robustness by incorporating gradient sparsity regularization during training; and Ding et al. (2024b) introduced stochastic gating mechanisms across layers to reduce the Lipschitz constant and suppress error amplification, ultimately strengthening SNN robustness. Summarily, existing methods primarily focus on constraining neuronal dynamics or regularization.

## 3 PRELIMINARIES

**Dynamics of leaky integrate-and-fire neuron.** In the field of SNNs, the most widely used neuron model is the Leaky Integrate-and-Fire (LIF), which simulates the charging and discharging process

of biological neurons to achieve activation (Fang et al., 2021). From an implementation perspective, the dynamics of a single LIF neuron can be expressed as follows:

$$
\begin{cases}
u[t+1] = (1 - \dfrac{1}{\tau})\big(u[t] - u_{\text{th}}s[t]\big) + c[t], \\
s[t+1] = \mathcal{H}\big(u[t+1] - u_{\text{th}}\big),
\end{cases}
\tag{1}
$$

where $u[t]$, $s[t]$, and $c[t]$ denote the membrane potential, the binary spike output, and the input current at time step $t$, respectively. Here, $\tau > 1$ is the membrane time constant controlling the leak rate, $V_{\text{th}}$ is the firing threshold, and $\mathcal{H}$ represents the Heaviside function. Training in SNNs is performed using BPTT combined with surrogate gradients, with detailed formulations of the BPTT gradients and the specific surrogate gradient methods provided in Appendix A.

**Adversarial attacks.** Given an input $x$ with label $y$, adversarial examples are generated by finding a perturbation $\delta$ within an $\ell_p$-norm ball of radius $\varepsilon$ that maximizes the loss $\mathcal{L}\big(f(x + \delta), y\big)$, that is,

$$
\underset{\|\delta\|_p \leq \varepsilon}{\arg\max} \ \mathcal{L}\big(f(x + \delta), y\big).
\tag{2}
$$

## 4 METHOD

In this section, we use empirical observations as the motivation and further introduce our method. Specifically, we first define gradient temporal collinearity as a measure of the alignment between gradient components across time steps. We then compare gradient temporal collinearity under direct encoding and rate encoding and analyze its impact on network robustness (Sec. 4.1). Based on the observed gap in gradient temporal collinearity between different encoding methods, we are inspired to develop our method: Structured Temporal Orthogonal Decorrelation (Sec. 4.2).

### 4.1 RATE VS. DIRECT: GRADIENT TEMPORAL COLLINEARITY ANALYSIS

While direct encoding substantially outperforms rate encoding in terms of efficiency and accuracy, its weakness in robustness is equally evident (Kundu et al., 2021; Kim et al., 2022; Mukhoty et al., 2025). This naturally raises a central question:

> 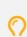 **Can the feature decorrelation mechanism inherent to rate encoding be borrowed to enhance the robustness of direct encoding without sacrificing its efficiency?**

Addressing this question requires more than empirical comparison; it calls for a principled metric that can systematically characterize the robustness gap. From an optimization perspective, robustness is closely tied to the spectral radius of the Hessian, which is in turn governed by the temporal structure of gradients. Motivated by this connection, we introduce Gradient Temporal Collinearity—a new measure that quantifies the directional consistency of gradient components across time steps, providing a rigorous lens through which the robustness of direct encoding can be analyzed and improved. We begin by defining:

**Definition 1.** *(Gradient Temporal Collinearity (GTC)). Let the network parameters be denoted as* $\theta = \{W^1, W^2, \ldots, W^L\}$. *The gradient* $\nabla_\theta \mathcal{L}$ *can be expressed, according to Eq. (10), as the sum of its $T$ components, that is,* $\nabla_\theta \mathcal{L} = \sum_{t=1}^{T} G[t]$, *where $G[t]$ is referred to as the component of the gradient at time step $t$. For any two gradient components $G[i], G[j]$ with $i \neq j$, we define their collinearity, i.e. gradient temporal collinearity $\mathcal{C}$, as in Eq. (3), where $\langle A, B \rangle_F = \text{Tr}(A^\top B)$ and $\|A\|_F = \sqrt{\langle A, A \rangle_F}$ denote the Frobenius inner product and Frobenius norm, respectively. The value $\mathcal{C} \in [-1, 1]$ quantifies the degree of collinearity between the two matrices: as $\mathcal{C} \to 1$, the two matrices become increasingly collinear, while smaller values indicate weaker collinearity.*

$$
\mathcal{C}(G[i], G[j]) = \frac{\langle G[i], G[j] \rangle_F}{\|G[i]\|_F \cdot \|G[j]\|_F},
\tag{3}
$$

**Definition 2.** *(Batch-Averaged and Epoch-Averaged GTC.) For the $b$-th batch, we define the batch-averaged GTC, denoted as $\overline{\mathcal{C}^b}$, as the mean collinearity among all pairs of gradient components within that batch. Based on this, the epoch-averaged GTC $\overline{\mathcal{C}}$ is defined as the mean of the batch-averaged*

*GTC values across all $B$ batches within an epoch. Both of these can be referred to Eq. (4), where $G^b[i]$ denotes the gradient component at time step $i$ in the $b$-th batch.*

$$\overline{\mathcal{C}^b} = \frac{2}{T(T-1)} \sum_{1 \le i < j \le T} \mathcal{C}\left(G^b[i], G^b[j]\right), \quad \overline{\mathcal{C}} = \frac{1}{B} \sum_{b=1}^{B} \overline{\mathcal{C}^b}. \tag{4}$$

**Experimental evaluation of GTC.** Next, we evaluate the evolution of GTC during training under the two encoding methods by varying the number of time steps across different datasets. The detailed setup for this experiment is provided in Appendix D.1.

As shown in Fig. 1, under different time step settings $T = 4, 8, 32$, the epoch-averaged GTC of direct encoding remains consistently higher than that of rate encoding. For direct encoding, the GTC starts around 0.8–0.9 in the early stage of training and gradually decreases while remaining relatively stable overall. In contrast, the GTC of rate encoding stays in a lower range of 0.2–0.3, with much larger fluctuations across epochs and without a clear decreasing trend during training. Moreover, within each encoding method, the trend of GTC variation does not

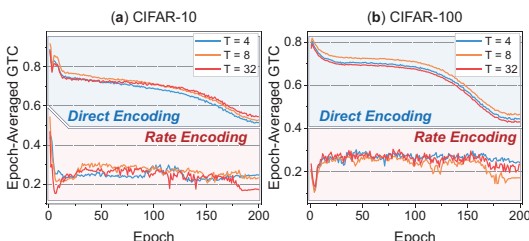

Figure 1: GTC evaluation curves.

depend on the number of time steps, and the overall magnitude of GTC shows no clear positive or negative correlation with the time step setting.

**Impact of GTC on SNN robustness.** While the above experiment highlights a clear difference in GTC between encoding methods, it remains to be explained why such a difference is crucial for SNN robustness. We address this by analyzing the spectral radius of the parameter Hessian, since robustness is closely tied to its magnitude (Stutz et al., 2021). The key result of our analysis (see Appendix B for the full derivation) is the following structured upper bound:

$$\lambda_{\max}(\widehat{H}_\theta) \lesssim T \cdot \left( \max_t \|G[t]\|_F^2 \right) \cdot \left[ 1 + (T-1) \cdot \max_{i \ne j} \mathcal{C}(G[i], G[j]) \right], \tag{5}$$

where $\widehat{H}_\theta$ is the parameter Hessian for a single sample and $\lambda_{\max}(\widehat{H}_\theta)$ indicates the spectral radius of it. This inequality shows that a higher of GTC directly amplifies the Hessian spectral radius, thereby reducing robustness.

**Origins of GTC disparities between encoding methods.** Having established that GTC affects robustness through its impact on the Hessian spectral radius, we next turn to the question of why direct encoding consistently exhibits higher GTC than rate encoding. The key lies in the temporal structure of the inputs. In direct encoding, every time step receives exactly the same input, leaving little temporal diversity. This produces highly consistent neuronal activations and gradient directions across time, leading to a near low-rank structure and consequently higher and more stable GTC. In contrast, rate encoding introduces stochastic spike sampling at the input, which serves as an inherent decorrelation mechanism across time steps. This reduces temporal collinearity of gradient components and thus yields much lower GTC, albeit with larger fluctuations across epochs due to the randomness involved. Moreover, from Fig. 1, we observe that the GTC curve of direct encoding gradually decreases as training progresses. This indicates that the strengthening robustness of direct encoding during training is accompanied by a process of gradient component dispersion, further highlighting the role of GTC in explaining the robustness gap between the two encoding methods.

## 4.2 STOD: Structurally Breaking Gradient Temporal Collinearity

Inspired by the inherent input feature decorrelation mechanism in rate encoding that enhances robustness, we propose Structured Temporal Orthogonal Decorrelation (STOD), a strategy that embeds feature decorrelation into the temporal dimension of direct encoding. The objective of STOD is to reduce GTC while preserving as much of direct encoding's feature-retention capability as possible, thereby achieving a balance between decorrelation and retention and ultimately enhancing the robustness of SNNs. STOD is composed of two key components: **Patchwise Feature Diversification** and **Global Orthogonal Regularization**, which we describe in detail in the following subsections.

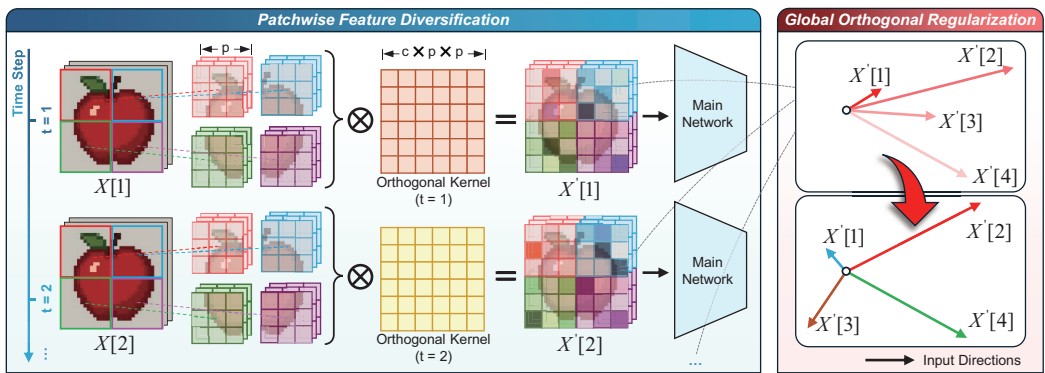

Figure 2: Flowchart of STOD, including PFD and GOR.

### 4.2.1 PATCHWISE FEATURE DIVERSIFICATION

The core idea of our method is to break the GTC, encouraging directional diversity across time steps so that external perturbations are misaligned in gradient space and their cumulative effect is mitigated. Since Sec. 4.1 shows that the high GTC of direct encoding originates from repeatedly injecting identical inputs across time while the low GTC of rate encoding arises from its inherent temporal diversity, a natural way to reduce GTC is to introduce controlled diversity into the temporal inputs of direct encoding. A seemingly intuitive approach would be to inject random noise into the inputs or gradients to disrupt this repetition, but such noise lacks mechanism awareness and offers no guarantee of producing meaningful temporal diversity; moreover, these artificial perturbations do not correspond to the actual variations encountered by the network and may instead cause gradient obfuscation (Athalye et al., 2018) and reduce explainability. To achieve this reduction in an explainable and stable manner, we first require the temporal inputs to exhibit well-controlled and systematically varied feature directions, rather than random or unstructured perturbations, so that the resulting gradient components naturally become less aligned across time. Building on this requirement, we introduce a structured input transformation module, Patchwise Feature Diversification (PFD), which applies, at each time step, an individual parametric orthogonal kernel to transform the input features.

However, as Fig. 2, directly applying transformations to the entire input is computationally expensive and renders optimization unstable. PFD first partitions the input into smaller, non-overlapping patches, enabling localized transformations while reducing complexity. Specifically, take static image data as an example, the input dimension is $X[t] \in \mathbb{R}^{C \times H \times W}$, and we define a partition operator $\mathcal{P}$ that divides the input into $N$ patches, that is $\mathcal{P}(X[t]) = \{X^n[t]\}_{n=1}^N$, with any $X^n[t] \in \mathbb{R}^{C \times p^2}$ and $N = \frac{H}{p} \cdot \frac{W}{p}$, where $p$ is a hyperparameter representing the patch size. A detailed discussion on the choice of $p$ is provided by ablation study (Sec. 5.2).

In an SNN with $T$ time steps, we denote the parametric orthogonal kernels as $\{\mathcal{O}[1], \mathcal{O}[2], \ldots, \mathcal{O}[T]\}$, where each $\mathcal{O}[t] \in \mathbb{R}^{d \times d}$ with $d = C \times p^2$. For any time step $t$, the original input $X[t]$ is mapped to the transformed input $X^n[t] \in \mathbb{R}^{CHW}$ as Eq. (6), where '$\otimes$' indicates the Kronecker product.

$$X'[t] = \text{vec}\big[\mathcal{P}^{-1}\big(\mathcal{P}(X[t]) \otimes \mathcal{O}[t]\big)\big]. \tag{6}$$

In addition, the orthogonal kernels are subject to the following structured constraints, ensuring both feature diversity and effective learning:

**Structured Constraint 1: Identity initialization at $t = 1$.** The first kernel is initialized as the identity matrix, i.e. $\mathcal{O}[1] = I_d$. This serves as a stable reference point, preventing all temporal inputs from being simultaneously distorted at initialization and ensuring that a portion of the original information is consistently preserved throughout training.

**Structured Constraint 2: Mutual orthogonality at initialization.** To maximize inter-step feature diversity from the outset and to avoid instability caused by overlapping gradient components during early training, all orthogonal kernels are initialized to be mutually orthogonal. Concretely, we adopt Householder reflections (Mhammedi et al., 2017) to enforce this constraint. Let $e_1, ..., e_d$ denote the canonical basis of $\mathbb{R}^d$. For steps $t > 1$, we construct vectors $k_j = e_1 - e_j$, for $j = 2, 3, ..., T$

(we use the first $T$ canonical basis vectors since $T \leq d$ in all settings) and define the corresponding Householder matrices as Eq. 7, then, the initialization set is given by $\{\mathcal{O}[1], \mathcal{O}[2], \ldots, \mathcal{O}[T]\} = \{I_d, Q[2], \ldots, Q[T]\}$.

$$Q[j] = I_d - 2\frac{k_j k_j^\top}{k_j^\top k_j}. \tag{7}$$

**Structured Constraint 3: Self-orthogonality during training.** During training, each orthogonal kernel is constrained to remain orthogonal, i.e., $\mathcal{O}[t]\mathcal{O}[t]^\top = I_d$. This ensures that the feature transformation at each time step preserves the energy structure of the input, namely $L_2$-norm conservation: $\|X'[t]\|_2 = \|X\|_2$. As a result, only the directional structure of the features is altered, avoiding meaningless scaling distortions or information loss. At the same time, this constraint preserves discriminative information and prevents pixel-intensity drift, since even slight intensity shifts at the pixel level may lead to entirely opposite network predictions (Williams & Li, 2023). In code practice, we register each orthogonal kernel as a `ManifoldParameter` constrained to the Stiefel manifold, and update them using a RiemannianSGD optimizer (Kochurov et al., 2020).

### 4.2.2 GLOBAL ORTHOGONAL REGULARIZATION

Ideally, the orthogonal kernels should also satisfy a fourth structured constraint: maintaining mutual orthogonality throughout training, rather than only at initialization. This would ensure persistent feature diversity across time steps and prevent temporal inputs from converging during optimization, which would otherwise reintroduce gradient collinearity. However, enforcing both mutual orthogonality and self-orthogonality as hard constraints simultaneously would make the system overly rigid, reducing the flexibility of parameter updates and ultimately hindering training effectiveness. Moreover, maintaining inter-kernel orthogonality during training requires concatenating all kernels and constraining them on a Stiefel manifold of dimension $d^2T$, which incurs prohibitive computational and memory costs, rendering this constraint impractical as a hard constraint.

Therefore, we propose Global Orthogonal Regularization (GOR), as Fig. 2 shows, as a soft constraint to achieve this effect. The objective of GOR is to guide the transformed inputs produced by the orthogonal kernels toward more diverse directions. We define GOR as:

$$\mathcal{L}_{\mathcal{O}} = \sum_{1 \leq i < j \leq T} \cos^2\left(\hat{X'}[i], \hat{X'}[j]\right), \tag{8}$$

where $\hat{X}'_i$ denotes the normalized transformed input. Thus, our final training objective is

$$\mathcal{L} = \mathcal{L}_{CE} + \lambda_{\mathcal{O}}\mathcal{L}_{\mathcal{O}}, \tag{9}$$

where $\mathcal{L}_{CE}$ is cross-entropy loss and $\lambda_{\mathcal{O}}$ is a hyperparameter controlling the strength of the regularization. A detailed discussion of $\lambda_{\mathcal{O}}$ setting is provided in the ablation study (Sec. 5.2).

## 5 EXPERIMENT

In this section, we first compare the robustness of our method with state-of-the-art methods (Sec.5.1). We then conduct ablation studies to analyze the contributions of different components of our method (Sec. 5.2). Finally, we inspect our method for any instances of gradient obfuscation (Sec. 5.3).

### 5.1 COMPARISON WITH STATE-OF-THE-ART (SOTA) METHODS

We conduct experiments on static visual classification datasets, including the small-scale CIFAR-10 and CIFAR-100 (Krizhevsky et al., 2009) , as well as the large-scale ImageNet (Deng et al., 2009), and Dynamic Vision Sensor (DVS) datasets DVS-CIFAR10 (Li et al., 2017) and DVS-Gesture (Amir et al., 2017). Detailed descriptions of these datasets and all preprocessing procedures are provided in Appendix C. For attack settings, we adopt Fast Gradient Sign Method (FGSM) (Goodfellow et al., 2014) and Projected Gradient Descent method (PGD) (Madry et al., 2017), both with $\varepsilon = 8/255$. Unless otherwise specified, the number of iterations $K$ in PGD is set to 7. The hyperparameter settings of our method are detailed in Appendix D.2.

**White box attacks.** White box attacks serve as the most direct means of evaluating the robustness of a method. Under this setting, we select several SOTA methods, including adversarial training

(AT) (Kundu et al., 2021), DLIF (Ding et al., 2024a), HoSNN (Geng & Li, 2023), FEEL (Xu et al., 2024), and StoG (Ding et al., 2024b), as baselines for comparison. Since the role of structured orthogonal kernels is primarily to regularize training by diversifying temporal feature representations, their effect is largely embedded into the learned network parameters after optimization. Therefore, during inference we remove the orthogonal kernels to highlight the intrinsic robustness gained from training, while also ensuring efficiency and fairness in comparison with baseline methods. This inference-time variant is denoted as STOD without Orthogonal Kernel (STOD w.o. OK). As shown in Table 1, even without using orthogonal kernels at inference, STOD w.o. OK achieves the most robust and balanced performance across all three datasets. For example, on CIFAR-10, HoSNN performs well under FGSM with an accuracy of 54.76%, but its performance drops dramatically to 15.32% under PGD. Conversely, FEEL achieves a strong 28.35% under PGD, but only 44.96% under FGSM. In contrast, our STOD w.o. OK consistently outperforms all baselines under both attacks. While the robustness gains on ImageNet are smaller than those on CIFAR datasets, this is expected. ImageNet's higher-resolution inputs and the deeper backbone naturally produce richer and more diverse representations, yielding lower temporal redundancy. As a result, STOD has a narrower margin for improvement, though it still provides consistent robustness gains under this challenging setting. Moreover, STOD can be further combined with AT (Kundu et al., 2021) to achieve additional performance gains. The corresponding experimental results are provided in Appendix E. Our method likewise demonstrates significant superiority under AT over existing SOTA approaches.

Table 1: White box performance comparison. The highest accuracy in each column is highlighted in bold. '*' indicates self-implementation results.

| Method | CIFAR-10 | | | CIFAR-100 | | | ImageNet | | |
|---|---|---|---|---|---|---|---|---|---|
| | Clean | FGSM | PGD | Clean | FGSM | PGD | Clean | FGSM | PGD |
| SNN | **93.75** | 8.19 | 0.03 | 72.39 | 4.55 | 0.19 | **57.84** | 4.99 | 0.01 |
| AT*(Kundu et al., 2021) | 91.16 | 38.20 | 14.07 | 69.69 | 16.31 | 8.49 | 51.00 | 15.74 | 6.39 |
| DLIF (Ding et al., 2024a) | 92.22 | 13.24 | 0.09 | 70.79 | 6.95 | 0.08 | - | - | - |
| HoSNN (Geng & Li, 2023) | 92.43 | 54.76 | 15.32 | 71.98 | 13.48 | 0.19 | - | - | - |
| FEEL (Xu et al., 2024) | 93.29 | 44.96 | 28.35 | **73.79** | 9.60 | 2.04 | - | - | - |
| StoG (Ding et al., 2024b) | 91.64 | 16.22 | 0.28 | 70.44 | 8.27 | 0.49 | - | - | - |
| STOD w.o. OK (Ours) | 91.43 | **55.80** | **32.97** | 71.00 | **26.26** | **13.13** | 54.02 | **19.08** | **6.44** |

Admittedly, according to Table 1, the clean accuracy of STOD is slightly lower than that of baseline methods. This is because all baselines rely on direct encoding, which repeatedly injects identical clean inputs and therefore maximally preserves clean feature fidelity. In contrast, STOD introduces structured temporal decorrelation at the input stage, replacing repeated clean inputs with diversified representations, which unavoidably results in a moderate clean-accuracy drop. But, this reduction is small, and the substantial robustness gains provided by STOD outweigh this minor degradation.

Table 2: White box performance comparison with-/without orthogonal kernels.

| Dataset | OK | Clean | FGSM | PGD |
|---|---|---|---|---|
| CIFAR-10 | ✗ | 91.43 | 55.80 | 32.97 |
| | ✓ | 90.87 ▼0.56 | 59.16 ▲3.36 | 36.72 ▲3.75 |
| CIFAR-100 | ✗ | 71.00 | 26.26 | 13.13 |
| | ✓ | 70.69 ▼0.31 | 32.15 ▲5.89 | 15.02 ▲1.89 |
| ImageNet | ✗ | 54.02 | 19.08 | 6.44 |
| | ✓ | 53.57 ▼0.45 | 20.93 ▲1.85 | 8.01 ▲1.57 |

Table 3: Accuracy comparison under RGA attacks.

| Dataset | Method | White Box RGA | | Black Box RGA | |
|---|---|---|---|---|---|
| | | FGSM | PGD | FGSM | PGD |
| CIFAR-10 | SNN | 8.01 | 1.01 | 29.80 | 13.09 |
| | STOD | 62.17 | 49.40 | 80.77 | 57.21 |
| CIFAR-100 | SNN | 6.89 | 0.64 | 18.94 | 9.20 |
| | STOD | 34.15 | 29.80 | 45.81 | 36.29 |

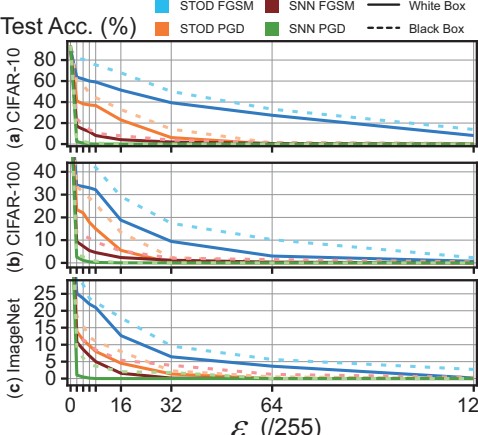

Figure 3: Performance comparison under white and black box attacks.

**Orthogonal kernels further improve SNN robustness.** While removing the orthogonal kernels at inference demonstrates that the robustness gains are intrinsically preserved in the trained parameters, retaining them provides an additional option to actively maintain structural decorrelation during deployment. When orthogonal kernels are enabled at inference (denoted STOD), temporal features remain explicitly decorrelated, leading to further robustness improvements under various white box attacks. As shown in Table 2, STOD consistently outperforms STOD w.o. OK under both FGSM and PGD. Importantly, the additional cost remains marginal; for example, under the main hyperparameter setting $T = 4$, $p = 8$, STOD introduces only 0.15M extra parameters (Detailed in Appendix F).

**White box attack in DVS datasets.** In DVS datasets, we trained the model and performed inference under FGSM and PGD attacks by directly perturbing the preprocessed event frames, as implemented in (Liu et al., 2024b). It can be seen from Table 4 that our method also demonstrates excellent robustness when dealing with the DVS dataset, surpassing SOTA method SR. Our method shows the same trend on the DVS

Table 4: Performance comparion in DVS datasets (%). he highest accuracy in each column is highlighted in bold. SR: (Liu et al., 2024b)

| Methods | DVS-CIFAR10 | | | DVS-Gesture | | |
|---|---|---|---|---|---|---|
| | Clean | FGSM | PGD | Clean | FGSM | PGD |
| SNN | **76.30** | 17.20 | 5.00 | **95.49** | 39.24 | 9.72 |
| SR | 75.50 | 64.60 | 61.20 | - | - | - |
| STOD w.o. OK | 74.70 | 66.10 | 60.40 | 93.75 | 87.15 | 54.67 |
| STOD | 72.90 | **68.80** | **62.50** | 93.05 | **88.88** | **56.60** |

dataset as on the static image dataset, that is, the inference without OK can surpass SOTA, and if OK is added, the robustness can be further enhanced.

**Black box attacks.** For black box attacks, adversarial examples are generated using the substitute model. We compare the performance gap of STOD between black box and white box settings and its performance trend under varying attack strengths as Fig. 3, with detailed experimental data provided in Appendix E, Table 9. It can be observed that robustness under black box attacks consistently exceeds that under white box attacks, and STOD exhibits a clear advantage over the vanilla SNN.

**Rate gradient approximation (RGA) attack.** RGA (Bu et al., 2023) leverages inter-layer spike rates in SNNs to perform more targeted adversarial attacks. Moreover, RGA is an incremental approach that must be combined with FGSM and PGD to construct the attacks. We evaluate the robustness of STOD under RGA attacks. As shown in Table 3, the vanilla method collapses severely under different forms of RGA, whereas our method demonstrates strong resistance against all such attacks.

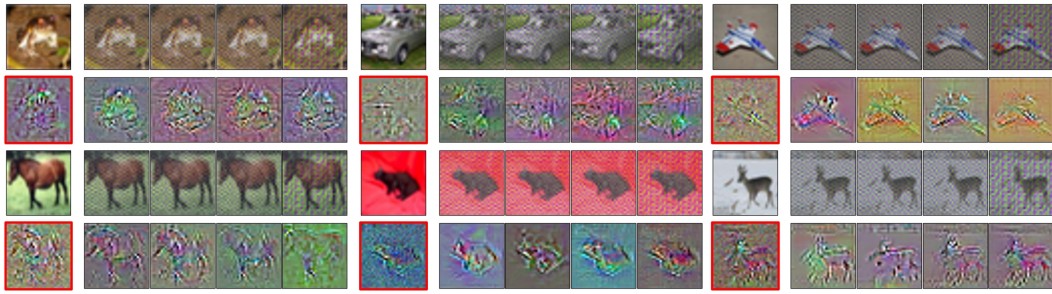

Figure 4: Visualization of original/transformed input and gradient components. For any given image, the top-left entry in the first row is the original input $X[t]$, while the next four images represent the transformed inputs $X'[t]$ obtained through PFD at $t = 1, 2, 3, 4$, respectively. In the second row, the top-left image within the red box shows the gradient component under direct encoding. Since the four gradient components of direct encoding are visually indistinguishable, we display only one of them. The remaining four images correspond to the gradient components ($G[t]$) of STOD at each time step.

**Visualization of transformed inputs and gradient components.** To further illustrate why STOD enhances network robustness, we visualize the input features at each time step and the corresponding gradient components (heatmap) in Fig. 4. Although the transformed inputs obtained by STOD are less visually discernible than the original images, its gradient components clearly reveal distinct structural patterns of the image from diverse perspectives. In contrast, the gradient components produced by direct encoding appear highly disordered and noisy, with almost no identifiable information to the human eye. This demonstrates that our method enhances the robustness of SNNs against adversarial attacks by leveraging orthogonal kernels to suppress gradient noise while diversifying gradients across time steps in an interpretable manner.

## 5.2 Ablation Study

In this subsection, we conduct ablation studies on the two most critical hyperparameters: the patch size $p$ and the regularization strength $\lambda_{\mathcal{O}}$.

**Patch size $p$.** On the CIFAR-10 dataset, we evaluate robustness with $T = 2, 4, 6, 8$ under different patch sizes $p = 2, 4, 8, 16, 32$. The detailed results are presented in Fig. 5. We observe that, across all time steps and regardless of whether orthogonal kernels are used at inference, STOD consistently achieves peak performance at $p = 8$. This indicates that overly small patches fail to capture spatial structure effectively, limiting temporal feature diversification,

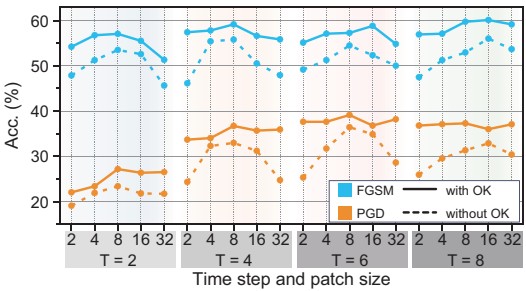

Figure 5: Accuracy comparison under different $p$. Detailed data provided in Appendix E, Table 10.

while overly large patches make the orthogonal transformation too coarse, discarding fine-grained local information. In contrast, medium-sized patches (e.g., 8) strike the best balance by preserving sufficient representational capacity while introducing an appropriate scale of perturbation, thereby achieving the optimal trade-off between temporal feature decorrelation and information fidelity.

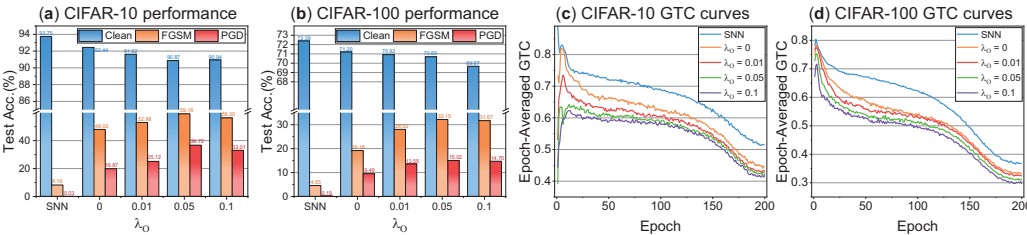

Figure 6: Performance and GTC curve comparison under different $\lambda_{\mathcal{O}}$.

**Regularization strength $\lambda_{\mathcal{O}}$.** We evaluate our method on CIFAR-10 and CIFAR-100 with $\lambda_{\mathcal{O}} = 0, 0.01, 0.05, 0.1$. As shown in Fig. 6, panels (a) and (b) compare robustness under these settings, while panels (c) and (d) illustrate the corresponding epoch-averaged GTC curves during training. The results show that STOD substantially reduces GTC throughout training, with stronger reductions observed as $\lambda_{\mathcal{O}}$ increases. However, greater regularization strength does not always lead to stronger robustness. The best performance is achieved when $\lambda_{\mathcal{O}} = 0.05$. It is important to note that STOD does not reduce GTC to the same extent as rate encoding. This discrepancy stems from the intrinsic nature of direct encoding: unlike rate encoding, which inherently randomizes temporal features, direct encoding must repeatedly preserve the original input to ensure feature fidelity. For this reason, aggressively reducing GTC to the level of rate encoding would inevitably erode essential information content. Instead, STOD deliberately seeks a principled balance—introducing sufficient temporal decorrelation to suppress error accumulation, while retaining the integrity of clean representations.

Beyond identifying the optimal values of each hyperparameter, the ablation results collectively indicate that STOD exhibits stable behavior rather than brittle sensitivity across a broad range of settings. For patch size $p$, performance varies smoothly as $p$ changes; suboptimal values (e.g., $p = 4$ or $p = 16$) still yield substantial robustness gains over the vanilla SNN, showing that STOD does not collapse when deviating from its optimum. A similar pattern emerges for the regularization strength $\lambda_{\mathcal{O}}$: although $\lambda_{\mathcal{O}} = 0.05$ performs best, all non-zero values consistently reduce GTC and improve robustness compared with the baseline, and no abrupt degradation is observed even when the regularization is moderately mis-tuned. This robustness plateau demonstrates that STOD benefits from a wide viable hyperparameter window, enabling strong generalization across datasets and architectures without precise tuning.

## 5.3 Inspection of Gradient Obfuscation

The seminal work Athalye et al. (2018) critically identified several characteristic behaviors, as summarized in Table 5, that arise when a defense attains spurious robustness through gradient

obfuscation. Accordingly, we evaluate DSD against each of these behaviors. Our experiments demonstrate that DSD passes all checks, as detailed follows: Table 5 summarizes the evaluation criteria. For items (1) and (2), it is clear from Fig. 3 and Table 3 that STOD performs significantly better under single-step attacks (FGSM) and black box attacks than under multi-step attacks (PGD) and white box attacks, respectively. For items (3) and (4), Fig. 3 shows that the performance of STOD drops to zero as the attack strength increases. Fig. 7 (with detailed data in Appendix E, Table 11) further confirms this trend, indicating that although the performance of STOD gradually degrades with more PGD iterations, it eventually converges to a stable minimum. Finally, item (5) states that gradient-based attacks fail to find adversarial examples; however, our results in Fig. 3 show the opposite—both FGSM and PGD consistently succeed in fooling STOD despite the training. In conclusion, our method does not utilize gradient obfuscation to achieve false robustness.

Table 5: Checklist for identifying gradient obfuscation.

| Characteristics to identify gradient obfuscation | Pass? |
| --- | --- |
| (1) Single-step attack performs better compared to iterative attacks | ✓ |
| (2) Black box attacks perform better compared to white box attacks | ✓ |
| (3) Increasing perturbation bound can't increase attack strength | ✓ |
| (4) Unbounded attacks can't reach 100% success | ✓ |
| (5) Adversarial example can be found through random sampling | ✓ |

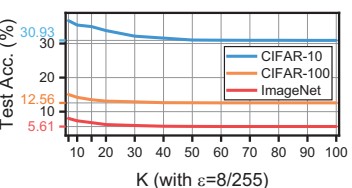

Figure 7: Acc. under different $K$.

## 6 CONCLUSION AND DISCUSSION

**Conclusion.** In this paper, we introduce the concept of Gradient Temporal Collinearity (GTC) and, from both experimental and theoretical perspectives, explain why rate encoding—with its lower GTC—exhibits stronger robustness than direct encoding, which suffers from higher GTC. Inspired by this phenomenon and its theoretical underpinnings, we borrow the feature decorrelation mechanism of rate encoding and structurally incorporate it into direct encoding, leading to our proposed method: Structured Temporal Orthogonal Decorrelation (STOD). STOD introduces parametric orthogonal kernels with structured constraints into the input layer to diversify input features, thereby breaking the high GTC of direct encoding and enhancing robustness. Extensive experiments demonstrate that our method consistently surpasses SOTA approaches in robustness, while comprehensive ablations confirm the effectiveness and necessity of each module. Together, these results position our method as a strong step toward more reliable and secure application of SNNs.

**Limitation.** Since our method reshapes network gradients to favor updates that enhance resistance to perturbations, it inevitably leads to a performance drop when the network is evaluated on clean data. This limitation is common across existing robustness-oriented studies in SNNs (Kundu et al., 2021; Ding et al., 2024a; Geng & Li, 2023; Xu et al., 2024; Ding et al., 2024b) and calls for deeper investigation to be effectively addressed.

**Broader impact.** When both inference modes: STOD with/without orthogonal kernels (OK) surpass existing SOTA methods, our approach offers practitioners greater flexibility. Users prioritizing maximum performance can opt for inference with OK, whereas those emphasizing efficiency and lightweight deployment may choose inference without OK. This dual advantage underscores the practical value of our method across diverse application scenarios.

## ACKNOWLEDGMENT

This work was supported in part by UK Research and Innovation (UKRI) Grant No. EP/Y036786/1, and Horizon Europe Grant No. 101129910.

## REPRODUCIBILITY STATEMENT

The complete code with fixed random seed utilized in this work is publicly available at `https://github.com/Apple26419/SNN_STOD`. All datasets employed in this research, including CIFAR-10, CIFAR-100, and ImageNet are publicly accessible. Details regarding the hardware,

coding environment, and hyperparameter settings used in our experiments are also included in the Appendix. We dedicate to enable future researchers to reproduce the results presented in this paper using similar computational setups.

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

## A    GRADIENT OF BPTT AND SURROGATE GRADIENT

In this section, we present the gradient formulation in the BPTT process, along with the surrogate gradient method employed.

In BPTT, let $W^l$ denote the weight matrix between layer $l$ and layer $l + 1$, and let $\mathcal{L}$ represent the loss function. The gradient can then be expressed as:

$$\frac{\partial \mathcal{L}}{\partial W^l} = \sum_{t=1}^{T} \frac{\partial \mathcal{L}}{\partial s^{l+1}[t]} \frac{\partial s^{l+1}[t]}{\partial u^{l+1}[t]} \left( \frac{\partial u^{l+1}[t]}{\partial W^l} + \sum_{k<t} \prod_{i=t-1}^{k} \left( \frac{\partial u^{l+1}[i+1]}{\partial u^{l+1}[i]} + \frac{\partial u^{l+1}[i+1]}{\partial s^{l+1}[i]} \frac{\partial s^{l+1}[i]}{\partial u^{l+1}[i]} \right) \frac{\partial u^{l+1}[k]}{\partial W^l} \right),$$

(10)

where the non-differentiable term $\frac{\partial s[t]}{\partial u[t]}$, we use the piecewise quadratic function (triangle function) as the surrogate gradient (Fang et al., 2023; Neftci et al., 2019), which is defined as:

$$g'(x) = \begin{cases} 0, & |x| > \frac{1}{\alpha} \\ -\alpha^2 |x| + \alpha, & |x| \leq \frac{1}{\alpha} \end{cases}.$$

(11)

The primitive function is defined as:

$$g(x) = \begin{cases} 0, & x < -\frac{1}{\alpha} \\ -\frac{1}{2}\alpha^2 |x|x + \alpha x + \frac{1}{2}, & |x| \leq \frac{1}{\alpha} \\ 1, & x > \frac{1}{\alpha} \end{cases},$$

(12)

where constant $\alpha$ is set to 1 for all cases.

## B    ANALYSIS OF GRADIENT TEMPORAL COLLINEARITY IMPACTS NETWORK ROBUSTNESS

In this section, we analyze what impact GTC will have on network robustness. This analysis is divided into two parts. In the first part, we analyze the impact of GTC on the spectral radius of the network parameter Hessian. In the second part, we analyze the influence of spectral radius on the robustness of the SNN network.

**Impact of GTC on Hessian Spectral Radius.** For a single sample $(x, y)$, let $z = f_\theta(x)$ be the network output and let the sample loss be $\ell(x, y; \theta)$. Given a batch $\mathcal{B} = \{(x_i, y_i)\}_{i=1}^{B}$, the overall objective is

$$\mathcal{L}(\theta) = \frac{1}{B} \sum_{i=1}^{B} \ell(x_i, y_i; \theta),$$

(13)

where $\theta$ is the set of network parameters as given by *Definition 1*.

We first recall the Gauss–Newton form of the parameter Hessian for a single sample $(x, y)$:

$$\widehat{H}_\theta(x, y; \theta) = J_\theta(x)^\top H_z(f_\theta(x); y) J_\theta(x),$$

(14)

where $J_\theta = \partial f_\theta(x)/\partial \theta$ is the parameter Jacobian, and $H_z = \partial^2 \ell(z, y)/\partial z^2$ is the Hessian of the loss $\ell(z, y)$ with respect to the network output $z = f_\theta(x)$. Then we analyze the temporal structure under BPTT. The network output can be written as the sum of contributions across time steps $f_\theta(x) = \sum_{t=1}^{T} f_\theta^{(t)}(x)$, and correspondingly we decompose

$$J_\theta = \sum_{t=1}^{T} J_t, \qquad J_t = \partial f_\theta^{(t)}(x)/\partial \theta,$$

(15)

so that the Gauss–Newton Hessian becomes

$$\widehat{H}_\theta = \sum_{t,t'} J_t^\top H_z J_{t'}.$$

(16)

Eq. (16) makes explicit that the Hessian contains both diagonal terms ($t = t'$) and cross-terms ($t \neq t'$), where the latter are directly determined by the temporal correlation of gradient components.

Next, we introduce a notational substitution $Z_t = H_z^{1/2} J_t$ to absorb the curvature weighting into each time block, so that

$$\widehat{H}_\theta = \Big(\sum_{t=1}^{T} Z_t\Big)^{\top} \Big(\sum_{t=1}^{T} Z_t\Big), \qquad \lambda_{\max}(\widehat{H}_\theta) = \Big\|\sum_{t=1}^{T} Z_t\Big\|_{op}^{2}. \tag{17}$$

We then define the temporal Gram matrix

$$\mathcal{T} \in \mathbb{R}^{T \times T}, \qquad \mathcal{T}_{ij} = \langle Z_i, Z_j \rangle_F, \tag{18}$$

so that $\|\sum_t Z_t\|_F^2 = \mathbf{1}^{\top} \mathcal{T} \mathbf{1}$, where $\mathbf{1}$ indicates vector with all elements of 1. Using the Rayleigh quotient, we have

$$\mathbf{1}^{\top} \mathcal{T} \mathbf{1} \leq T \lambda_{\max}(\mathcal{T}), \tag{19}$$

and therefore

$$\lambda_{\max}(\widehat{H}_\theta) \leq T \lambda_{\max}(\mathcal{T}). \tag{20}$$

After obtaining this bound, we continue to further bound $\lambda_{\max}(\mathcal{T})$. Let $M \in \mathbb{R}^{T \times T}$ denote the normalized correlation matrix with

$$M_{ij} = \frac{\langle Z_i, Z_j \rangle_F}{\|Z_i\|_F \|Z_j\|_F}, \quad M_{ii} = 1, \tag{21}$$

and let $D = \mathrm{diag}(\|Z_1\|_F, \ldots, \|Z_T\|_F)$ so that $\mathcal{T} = DMD$. Then by the submultiplicativity of the operator norm we have

$$\lambda_{\max}(\mathcal{T}) \leq \big(\max_t \|Z_t\|_F^2\big) \cdot \lambda_{\max}(M). \tag{22}$$

Applying Gershgorin's circle theorem, we obtain

$$\lambda_{\max}(M) \leq 1 + (T-1) \cdot \max_{i \neq j} |M_{ij}|. \tag{23}$$

Combining Eqs. (20), (22), and (23), we arrive at the structured upper bound

$$\lambda_{\max}(\widehat{H}_\theta) \leq T \cdot \Big(\max_t \|Z_t\|_F^2\Big) \cdot \Big[1 + (T-1) \cdot \max_{i \neq j} |M_{ij}|\Big]. \tag{24}$$

Finally, we may replace $M_{ij}$ by GTC $\mathcal{C}(G[i], G[j])$ and $\|Z_t\|_F^2$ by $\|G[t]\|_F^2$. Hence we obtain

$$\lambda_{\max}(\widehat{H}_\theta) \lesssim T \cdot \Big(\max_t \|G[t]\|_F^2\Big) \cdot \Big[1 + (T-1) \cdot \max_{i \neq j} \mathcal{C}(G[i], G[j])\Big]. \tag{25}$$

This structured bound makes explicit that the spectral radius of the parameter Hessian is determined jointly by the per-step gradient magnitudes and the temporal gradient similarities. In particular, when $\mathcal{C}(G[i], G[j])$ is close to 1 for all pairs, the gradients are nearly collinear across time steps, leading to a near rank-1 [1] structure that amplifies cross-terms and yields a quadratic growth of the spectral radius with $T$; when $\mathcal{C}(G[i], G[j])$ is small due to temporal decorrelation, the growth is closer to linear. Therefore, lowering the GTC directly reduces the structured upper bound of $\lambda_{\max}(\widehat{H}_\theta)$.

**Impact of the Hessian Spectral Radius on Network Robustness.** The paper (Stutz et al., 2021) presents from an experimental perspective the conclusion that the larger the spectral radius, the less robust the network. We conduct a brief theoretical analysis of this conclusion based on SNN.

From the perspective of a single sample, we analyze why an increase in the spectral radius of the network parameters amplifies the effect of input perturbations on the loss function, thereby indicating that the network's robustness deteriorates as the spectral radius grows.

Based on Eq. (13), for any input perturbation $\delta$ with $\|\delta\| \leq \varepsilon$, Taylor's theorem at $(x, y; \theta)$ gives

$$\ell(x + \delta, y; \theta) - \ell(x, y; \theta) = \nabla_x \ell(x, y; \theta)^{\top} \delta + \tfrac{1}{2} \delta^{\top} H_x(x, y; \theta) \delta + o(\|\delta\|^2), \tag{26}$$

where $H_x(x, y; \theta) = \nabla_x^2 \ell(x, y; \theta)$ is the input Hessian for the sample.

By the Rayleigh-quotient bound $\delta^{\top} H_x \delta \leq \lambda_{\max}(H_x)\|\delta\|^2$, where $\lambda_{\max}(\cdot)$ is the spectral radius for symmetric matrices, we obtain the sample-wise worst-case bound

$$\sup_{\|\delta\| \leq \varepsilon} \big(\ell(x + \delta, y; \theta) - \ell(x, y; \theta)\big) \leq \varepsilon \|\nabla_x \ell(x, y; \theta)\| + \tfrac{1}{2} \varepsilon^2 \lambda_{\max}(H_x(x, y; \theta)). \tag{27}$$

---

[1] Near rank-1 refers to the dominance of the leading eigenvalue of the temporal correlation matrix.

Next, we should further obtain the relation between input Hessian $H_x(x, y; \theta)$ and overall parameter Hessian $H_\theta$. Parameter Hessian can be expressed as

$$H_\theta(\theta) = \nabla_\theta^2 \mathcal{L}(\theta) = \frac{1}{B} \sum_{i=1}^{B} \widehat{H}_\theta(x_i, y_i; \theta), \tag{28}$$

where $\widehat{H}_\theta(x, y; \theta) = \nabla_\theta^2 \ell(x, y; \theta)$ is the parameter Hessian for a single sample.

First, the input Hessian admits the exact decomposition

$$H_x(x, y; \theta) = J_f(x)^\top H_z\big(f_\theta(x); y\big) J_f(x) + R(x, y; \theta), \tag{29}$$

where $H_z(f_\theta(x); y)$ is the output Hessian, and $J_f(x) = \partial f_\theta(x)/\partial x$ is the input Jacobian. In the exact decomposition, the residual term $R(x, y; \theta)$ arises from the interaction between the loss gradient $\nabla_z \ell(x, y; \theta)$ and the second derivatives $\partial^2 f_\theta(x)/\partial x^2$ of the network with respect to the input. While this term may be nonzero in general, near convergence the loss gradient becomes small, and thus its contribution can be upper-bounded by a constant $\eta \geq 0$, i.e., $\|R(x, y; \theta)\| \leq \eta$. For the cross-entropy loss, which is generally used in SNN fields, the softmax Hessian takes the form $\mathrm{diag}(p) - pp^\top$, which is positive semidefinite in the de-centered subspace orthogonal to the all-ones vector. All nonzero eigenvalues of this Hessian lie in the interval $[0, 1]$. Therefore, we can take $\beta = 1$ as an upper bound on the spectral norm of $H_z$, i.e., $\|H_z\| \leq \beta$, and proceed with the subsequent analysis restricted to this subspace.

On the other hand, for the input Hessian $H_x$, its positive semidefiniteness cannot be guaranteed in general. This is because the nonlinearity of the network may lead to non-convexity. Consequently, we can conclude that its spectral radius is bounded above by its spectral norm, namely,

$$\lambda_{\max}\big(H_x(x, y; \theta)\big) \leq \|H_x(x, y; \theta)\|. \tag{30}$$

By combining Eqs. (29) and (30) with the above analysis, we obtain the relation as follows:

$$\lambda_{\max}\big(H_x(x, y; \theta)\big) \leq \|H_x(x, y; \theta)\| \leq \|J_f(x)\|^2 \|H_z\| + \|R(x, y; \theta)\| \leq \beta \|J_f(x)\|^2 + \eta. \tag{31}$$

After obtaining the crucial inequality relation as Eq. (31), We further analyze the relationship between the input Jacobian $J_f(x) = \partial f_\theta(x)/\partial x$ and the parameter Jacobian $J_\theta(x) = \partial f_\theta(x)/\partial \theta$.

In a SNN with $L$-layer feedforward network over $T$ steps, the pre-activation is membrane potential. For the $l$-th layer at time $t$, we define

$$D_l[t] = \mathrm{diag}\big(\phi(u_l[t])\big), \quad \mathrm{U}_\phi = \sup_u \big|\phi(u)\big| < \infty, \tag{32}$$

where $\phi$ is surrogate gradient function, and $\mathrm{U}_\phi$ is the upper bound of surrogate gradient function, which is a bounded value for generally used surrogate gradient functions, such as triangle, rectangle, and sigmoid. Based on Eq. (32), we can use chain rule across layer and time steps after unrolling the network and obtain

$$J_f(x) = \prod_{t=1}^{T} \Big( W_L D_{L-1}[t] W_{L-1} D_{L-2}[t] \cdots D_1[t] W_1 \Big), \tag{33}$$

hence, by $\|D_l[t]\| \leq \mathrm{U}_\phi$, we get

$$\|J_f(x)\| \leq \Big( \mathrm{U}_\phi^{L-1} \prod_{l=1}^{L} \|W_l\| \Big)^T. \tag{34}$$

The parameter Jacobian $J_\theta(x)$ stacks the blocks $\partial f_\theta(x)/\partial \mathrm{vec}(W_l)$. Assuming $c_0(x) > 0$ is a non-degeneracy constant that aggregates lower bounds on intermediate activations and on surrogate slopes over active neurons across layers/time, such that

$$\|J_\theta(x)\| \geq c_0(x) \Big( \prod_{l=1}^{L} \|W_l\| \Big)^T. \tag{35}$$

Substituting Eq. (35) into Eq. (34) yields

$$\|J_f(x)\| \leq \mathcal{K}(x) \|J_\theta(x)\|, \tag{36}$$

with the input-dependent factor

$$\mathcal{K}(x) \;=\; \frac{\mathrm{U}_\phi^{L-1}}{c_0(x)}. \tag{37}$$

Next, we link $\|J_f\|$ to $H_\theta(\theta)$ via a structural inequality and Gauss-Newton method. On the parameter side, for the sample $(x, y)$ we have

$$\widehat{H}_\theta(x, y; \theta) \;=\; J_\theta(x)^\top H_z\big(f_\theta(x); y\big) J_\theta(x) \;+\; R_\theta(x, y; \theta), \tag{38}$$

with a higher-order residual $R_\theta$. In a Gauss-Newton regime, $\|R_\theta\|$ is negligible, and thus we can get Eq. (39), where '$\succeq$' is Loewner order.

$$\widehat{H}_\theta(x, y; \theta) \;\succeq\; J_\theta(x)^\top H_z\big(f_\theta(x); y\big) J_\theta(x). \tag{39}$$

Let $\mu_{\mathrm{eff}}$ be the minimal eigenvalue of $H_z$ on the working subspace. Then

$$\|J_\theta(x)\|^2 \;\leq\; \frac{1}{\mu_{\mathrm{eff}}} \lambda_{\max}\big(\widehat{H}_\theta(x, y; \theta)\big). \tag{40}$$

Since $H_\theta(\theta) = \frac{1}{B} \sum_{i=1}^B \widehat{H}_\theta(x_i, y_i; \theta)$ and each $\widehat{H}_\theta(x_i, y_i; \theta) \succeq 0$ near minima, we have

$$H_\theta(\theta) \;\succeq\; \frac{1}{B} \widehat{H}_\theta(x, y; \theta) \quad \Rightarrow \quad \lambda_{\max}\big(H_\theta(\theta)\big) \;\geq\; \frac{1}{B} \lambda_{\max}\big(\widehat{H}_\theta(x, y; \theta)\big). \tag{41}$$

Combining Eq. (40) and Eq. (41) yields

$$\|J_\theta(x)\|^2 \;\leq\; \frac{B}{\mu_{\mathrm{eff}}} \lambda_{\max}\big(H_\theta(\theta)\big). \tag{42}$$

Using Eq. (36) and Eq. (42),

$$\|J_f(x)\|^2 \;\leq\; \mathcal{K}(x)^2 \|J_\theta(x)\|^2 \;\leq\; \frac{B \, \mathcal{K}(x)^2}{\mu_{\mathrm{eff}}} \lambda_{\max}\big(H_\theta(\theta)\big). \tag{43}$$

Substituting Eq. (43) into Eq. (31) gives

$$\lambda_{\max}\big(H_x(x, y; \theta)\big) \;\leq\; \frac{\beta \, B \, \mathcal{K}(x)^2}{\mu_{\mathrm{eff}}} \lambda_{\max}\big(H_\theta(\theta)\big) \;+\; \eta. \tag{44}$$

Putting Eq. (44) into the perturbation bound Eq. (27), we obtain the general inequality

$$\sup_{\|\delta\| \leq \varepsilon} \big(\ell(x+\delta, y; \theta) - \ell(x, y; \theta)\big) \;\leq\; \varepsilon\|\nabla_x \ell(x, y; \theta)\| + \tfrac{1}{2}\varepsilon^2\Big(\frac{\beta \, B \, \mathcal{K}(x)^2}{\mu_{\mathrm{eff}}} \lambda_{\max}\big(H_\theta(\theta)\big) + \eta\Big). \tag{45}$$

At (near-)stationary points where $\|\nabla_x \ell(x, y; \theta)\| \approx 0$, the worst-case loss increase reduces to

$$\sup_{\|\delta\| \leq \varepsilon} \big(\ell(x+\delta, y; \theta) - \ell(x, y; \theta)\big) \;\leq\; \tfrac{1}{2}\varepsilon^2\Big(\frac{\beta \, B \, \mathcal{K}(x)^2}{\mu_{\mathrm{eff}}} \lambda_{\max}\big(H_\theta(\theta)\big) + \eta\Big), \tag{46}$$

Although the terms $\mu_{\mathrm{eff}}$ and $\mathcal{K}(x)$ in Eq. (46) are influenced by the network's input and output, they are not directly coupled with $\lambda_{\max}\big(H_\theta(\theta)\big)$. Hence, they only affect the scaling factor on the right-hand side of the equation and do not alter its monotonicity when treating $\lambda_{\max}\big(H_\theta(\theta)\big)$ as the variable. Therefore, we conclude that under a fixed perturbation strength $\varepsilon$, the upper bound of the network loss fluctuation increases with the spectral radius $\lambda_{\max}\big(H_\theta(\theta)\big)$. In other words, the robustness of the network is weakened as its spectral radius grows.

## C   DATASET

**CIFAR-10.** The CIFAR-10 dataset (Krizhevsky et al., 2009) consists of 60,000 color images, each of size 32×32 pixels, divided into 10 different classes, such as airplanes, cars, birds, cats, and dogs. Each class has 6,000 images, with 50,000 images used for training and 10,000 for testing. Normalization, random horizontal flipping, random cropping with 4 padding, and CutOut (DeVries & Taylor, 2017) are applied for data augmentation.

**CIFAR-100.** The CIFAR-100 dataset (Krizhevsky et al., 2009) consists of 60,000 color images, each of size 32×32 pixels, categorized into 100 different classes. Each class contains 600 images, with 500 used for training and 100 for testing. The same processing methods as for dataset CIFAR-10 are applied to dataset CIFAR-100.

**ImageNet.** We evaluate on the ILSVRC-2012 ImageNet dataset (Deng et al., 2009), which contains ∼1.28M training images and 50,000 validation images spanning 1,000 classes. Images are of variable resolution; following common practice and our implementation, training augmentation includes RandomResizedCrop to $224 \times 224$, RandomHorizontalFlip, conversion to tensors, and channel-wise normalization. For test, images are resized to have a shorter side of 256 pixels and then center-cropped to $224 \times 224$ before applying the same normalization.

**DVS-CIFAR10.** The DVS-CIFAR-10 dataset (Li et al., 2017) is a neuromorphic version of the traditional CIFAR-10 dataset. DVS-CIFAR10 captures the visual information using a Dynamic Vision Sensor (DVS), which records changes in the scene as a series of asynchronous events rather than as a sequence of frames. The dataset consists of recordings of 10 object classes, corresponding to the original CIFAR-10 categories, with each object presented in front of a DVS camera under various conditions. The dataset contains 10,000 128×128 images, of which 9,000 are used as the training set and the remaining 1,000 as the test set.

**DVS-Gesture.** The DVS-Gesture dataset (Amir et al., 2017) is a neuromorphic dataset, consisting of 11 different hand gesture classes, such as hand clapping, arm rolling, and air guitar, performed by 29 subjects under various lighting conditions. Each gesture is represented by a sequence of events rather than frames. The dataset contains 1,176 training samples and 288 testing samples.

# D EXPERIMENTAL SETUP

## D.1 EXPERIMENTAL SETUP FOR GTC EVALUATION

In the GTC evaluation experiments, we consider both direct encoding and rate encoding. In direct encoding, identical input data are fed into the network at every time step. For rate encoding, we adopt Poisson encoding (Lee et al., 2020), where input pixel values are converted into Poisson-distributed spike trains and presented to the network. Typically, direct encoding requires fewer time steps (e.g., 4 or 8), whereas rate encoding requires more (e.g., 32). To ensure a rigorous comparison, we evaluate both encoding methods under time steps $T = 4, 8, 32$. For both methods, the data preprocessing follows Appendix C, and the hyperparameters used for evaluation are listed in the following table. All other experimental settings not mentioned are identical to those described in Appendix D.2.

Table 6: Hyperparameter settings for GTC evaluation experiments.

| Dataset | Optimizer | Model | LeaningRate | WeightDecay | Epoch | BatchSize |
|---------|-----------|-------|-------------|-------------|-------|-----------|
| CIFAR-10 | SGD | VGG-11 | 0.1 | 5e-5 | 200 | 128 |
| CIFAR-100 | SGD | VGG-11 | 0.1 | 5e-4 | 200 | 128 |

## D.2 EXPERIMENTAL SETUP FOR MAIN EXPERIMENTS

In our experiments, all training cases are implemented using PyTorch (Paszke et al., 2019) with the SpikingJelly (Fang et al., 2023) framework and executed on an NVIDIA GeForce RTX 5090 GPU. For each dataset, we utilize the hyperparameters listed as the following table, consistently employing the SGD optimizer for network parameters and RiemannianSGD optimizer for orthogonal kernels and the membrane time constant $\tau$ to 1.1. Unless otherwise specified, the ablation studies are conducted under the same experimental settings as the main experiments.

# E DETAILED EXPERIMENTAL RESULTS

This section first presents a comparison of our method with SOTA approaches under AT in terms of robustness against white box attacks (Table 8). We then provide the complete experimental results used to generate Fig. 3 (Table 9), Fig. 5 (Table 10), and Fig. 7 (Table 11) in the main text.

Table 7: Hyperparameter settings for main experiments. 'LR' denotes the learning rate, 'LRO' is the learning rate for orthogonal kernels. NF-ResNet-18* (Brock et al., 2021).

| Dataset | Model | Timestep | LR | LRO | WeightDecay | Epoch | BatchSize | $p$ | $\lambda_O$ |
|---------|-------|----------|-----|-----|-------------|-------|-----------|-----|-------------|
| CIFAR-10 | VGG-11 | 4 | 0.1 | 0.05 | 5e-5 | 200 | 128 | 8 | 0.05 |
| CIFAR-100 | VGG-11 | 4 | 0.1 | 0.05 | 5e-4 | 200 | 128 | 8 | 0.05 |
| ImageNet | NF-ResNet-18* | 4 | 0.1 | 0.05 | 1e-5 | 150 | 512 | 16 | 0.05 |
| DVS-CIFAR10 | VGG-11 | 10 | 0.05 | 0.05 | 5e-4 | 200 | 128 | 8 | 0.05 |
| DVS-Gesture | VGG-11 | 20 | 0.05 | 0.05 | 5e-5 | 200 | 8 | 8 | 0.05 |

Table 8: White box performance (with AT) comparison. The highest accuracy in each column is highlighted in bold. '*' indicates self-implementation results.

| Method | CIFAR-10 | | | CIFAR-100 | | | ImageNet | | |
|--------|----------|------|-----|-----------|------|-----|----------|------|-----|
| | Clean | FGSM | PGD | Clean | FGSM | PGD | Clean | FGSM | PGD |
| SNN + AT* (Kundu et al., 2021) | **91.16** | 38.20 | 14.07 | 69.69 | 16.31 | 8.49 | 51.00 | 15.74 | 6.39 |
| DLIF + AT (Ding et al., 2024a) | 88.91 | 56.71 | 40.30 | 66.33 | 36.83 | 24.25 | - | - | - |
| HoSNN + AT (Geng & Li, 2023) | 90.00 | 63.98 | 43.33 | 64.64 | 26.97 | 16.66 | - | - | - |
| FEEL + AT (Xu et al., 2024) | - | - | - | **69.79** | 18.67 | 11.07 | - | - | - |
| StoG + AT (Ding et al., 2024b) | 90.13 | 45.74 | 27.74 | 66.37 | 24.45 | 14.42 | - | - | - |
| STOD + AT w.o. OK (Ours) | 89.70 | 68.79 | 39.99 | 68.02 | 39.76 | 24.31 | **51.92** | 23.86 | 8.43 |
| STOD + AT (Ours) | 88.94 | **73.23** | **43.54** | 67.19 | **41.89** | **27.94** | 50.68 | **26.77** | **9.80** |

## F    ADDITIONAL PARAMETER NUMBER ASSESSMENT

**It is important to note that the orthogonal kernels in our method can be discarded during inference, ensuring that no additional inference overhead is introduced.** Specifically, in Table 1 of the main experiments, we compare STOD without orthogonal kernels (STOD w.o. OK) against other SOTA methods. In Table 2, we further report the performance gap between inference with and without orthogonal kernels.

In this section, we calculate the additional number of parameters introduced by the orthogonal kernels. Given that the kernel dimension is $d = C \times p \times p$, in an SNN with $T$ time steps, the extra parameters $N_{\text{para}}$ can be computed as:

$$N_{\text{para}} = T(Cp^2)^2 \tag{47}$$

Therefore, when $C = 3$ and $T = 4$, if we set $p = 2, 4, 8, 16, 32$, the additional number of parameter would be 576, 9216, 0.15M, 2.36M, and 37.75M, respectively. As shown, when $p = 8$, only 0.15M additional parameters are introduced, which is nearly negligible.

## G    STATEMENT OF LARGE LANGUAGE MODEL (LLM) USAGE

In the preparation of this manuscript, an LLM was employed to assist with non-scientific tasks. These included polishing the English writing for clarity and style, providing suggestions for figure design and color schemes, supporting LaTeX formatting and typesetting, and drafting this statement.

Table 9: Performance of STOD with different attack methods. WB and BB denote white and black box, respectively. This is detailed experimental results of Fig. 3.

| Attack | $\varepsilon = 0$ | 2 | 4 | 6 | 8 | 16 | 32 | 64 | 128 |
|---|---|---|---|---|---|---|---|---|---|
| **CIFAR-10** | | | | | | | | | |
| SNN FGSM WB | 93.75 | 17.06 | 14.22 | 11.78 | 8.19 | 4.21 | 1.99 | 0.58 | 0.00 |
| SNN FGSM BB | 93.75 | 24.13 | 18.41 | 13.86 | 10.26 | 7.88 | 3.82 | 1.48 | 0.59 |
| SNN PGD WB | 93.75 | 2.37 | 1.01 | 0.34 | 0.03 | 0.00 | 0.00 | 0.00 | 0.00 |
| SNN PGD BB | 93.75 | 4.01 | 2.80 | 1.20 | 0.89 | 0.02 | 0.00 | 0.00 | 0.00 |
| STOD FGSM WB | 90.87 | 63.71 | 62.08 | 60.04 | 59.16 | 51.49 | 39.45 | 27.49 | 8.17 |
| STOD FGSM BB | 90.87 | 84.36 | 80.88 | 77.74 | 75.42 | 67.91 | 50.09 | 33.18 | 13.95 |
| STOD PGD WB | 90.87 | 41.10 | 38.89 | 37.29 | 36.72 | 23.07 | 6.19 | 0.04 | 0.00 |
| STOD PGD BB | 90.87 | 60.42 | 56.29 | 48.70 | 44.24 | 32.96 | 14.31 | 1.07 | 0.51 |
| **CIFAR-100** | | | | | | | | | |
| SNN FGSM WB | 72.39 | 9.37 | 7.42 | 5.46 | 4.55 | 2.35 | 1.19 | 0.33 | 0.00 |
| SNN FGSM BB | 72.39 | 13.26 | 12.11 | 10.84 | 9.16 | 5.31 | 2.07 | 1.47 | 0.49 |
| SNN PGD WB | 72.39 | 2.53 | 1.15 | 0.50 | 0.19 | 0.02 | 0.00 | 0.00 | 0.00 |
| SNN PGD BB | 72.39 | 3.65 | 2.68 | 1.89 | 0.78 | 0.15 | 0.02 | 0.00 | 0.00 |
| STOD FGSM WB | 70.69 | 34.15 | 33.56 | 32.99 | 32.15 | 18.79 | 9.44 | 3.00 | 0.71 |
| STOD FGSM BB | 70.69 | 49.44 | 46.30 | 44.25 | 41.96 | 29.22 | 17.40 | 10.22 | 2.27 |
| STOD PGD WB | 70.69 | 23.42 | 20.00 | 17.91 | 15.02 | 5.55 | 0.02 | 0.00 | 0.00 |
| STOD PGD BB | 70.69 | 33.60 | 31.08 | 28.58 | 25.90 | 13.46 | 2.40 | 0.46 | 0.00 |
| **ImageNet** | | | | | | | | | |
| SNN FGSM WB | 57.84 | 10.75 | 8.59 | 6.73 | 4.99 | 1.56 | 0.14 | 0.01 | 0.00 |
| SNN FGSM BB | 57.84 | 12.13 | 11.43 | 10.25 | 8.46 | 5.42 | 3.93 | 1.25 | 0.35 |
| SNN PGD WB | 57.84 | 1.02 | 0.48 | 0.13 | 0.01 | 0.00 | 0.00 | 0.00 | 0.00 |
| SNN PGD BB | 57.84 | 9.19 | 6.05 | 4.06 | 3.67 | 2.02 | 1.79 | 0.05 | 0.00 |
| STOD FGSM WB | 53.57 | 25.10 | 23.65 | 21.99 | 20.93 | 12.68 | 6.43 | 3.65 | 0.17 |
| STOD FGSM BB | 53.57 | 30.59 | 27.75 | 23.64 | 22.50 | 17.94 | 9.61 | 5.64 | 2.70 |
| STOD PGD WB | 53.57 | 13.86 | 11.38 | 9.71 | 8.01 | 4.52 | 1.36 | 0.02 | 0.00 |
| STOD PGD BB | 53.57 | 18.43 | 15.69 | 12.23 | 10.98 | 7.98 | 2.29 | 0.51 | 0.02 |

Table 10: Performance of STOD with different time steps and patch sizes. This is detailed experimental results of Fig. 5.

| Attack | OK | T = 2 | | | | | T = 4 | | | | | T = 6 | | | | | T = 8 | | | | |
|---|---|---|---|---|---|---|---|---|---|---|---|---|---|---|---|---|---|---|---|---|---|
| | | p=2 | 4 | 8 | 16 | 32 | 2 | 4 | 8 | 16 | 32 | 2 | 4 | 8 | 16 | 32 | 2 | 4 | 8 | 16 | 32 |
| FGSM | ✗ | 47.88 | 51.26 | 53.51 | 52.58 | 45.65 | 46.14 | 55.41 | 55.80 | 50.54 | 47.94 | 49.19 | 51.27 | 54.49 | 52.35 | 50.02 | 47.50 | 51.26 | 52.95 | 56.01 | 53.69 |
| | ✓ | 54.22 | 56.77 | 57.06 | 55.52 | 51.33 | 57.43 | 57.82 | 59.16 | 56.62 | 55.84 | 55.14 | 57.10 | 57.29 | 58.82 | 54.83 | 56.93 | 57.10 | 59.74 | 60.12 | 59.16 |
| PGD | ✗ | 19.09 | 21.92 | 23.40 | 21.81 | 21.72 | 24.33 | 32.30 | 32.97 | 31.21 | 24.75 | 25.32 | 31.73 | 36.45 | 34.89 | 28.63 | 25.95 | 29.56 | 31.34 | 32.89 | 30.40 |
| | ✓ | 22.05 | 23.40 | 27.19 | 26.37 | 26.52 | 33.69 | 34.05 | 36.72 | 35.69 | 35.92 | 37.63 | 37.65 | 39.14 | 36.80 | 38.19 | 36.80 | 37.11 | 37.30 | 36.02 | 37.08 |

Table 11: Performance of STOD with different PGD step number. This is detailed experimental results of Fig. 7.

| Dataset | $K = 7$ | 10 | 15 | 20 | 30 | 40 | 50 | 60 | 70 | 80 | 90 | 100 |
|---|---|---|---|---|---|---|---|---|---|---|---|---|
| CIFAR-10 | 36.72 | 35.44 | 34.98 | 33.80 | 32.17 | 31.59 | 31.02 | 30.97 | 30.95 | 30.94 | 30.93 | 30.93 |
| CIFAR-100 | 15.02 | 14.20 | 13.50 | 13.09 | 12.87 | 12.62 | 12.59 | 12.57 | 12.56 | 12.56 | 12.56 | 12.56 |
| ImageNet | 8.01 | 7.29 | 6.77 | 6.20 | 5.92 | 5.71 | 5.65 | 5.63 | 5.62 | 5.62 | 5.62 | 5.61 |

