# OpenReview forum: "Breaking Gradient Temporal Collinearity for Robust Spiking Neural Networks"
_ICLR.cc/2026/Conference — ICLR 2026 Poster_

### Official Review · Reviewer_6hhX · 2025-10-28

**Soundness:** 2
**Presentation:** 3
**Contribution:** 2
**Rating:** 4
**Confidence:** 3

**Summary:**

This paper aims to improve the robustness of spiking neural networks (SNNs) under direct coding. The authors first proposes a metric called Gradient Temporal Collinearity (GTC) to measure robustness by how similar gradients are across time. Then the authors propose Structured Temporal Orthogonal Decorrelation (STOD), which applies orthogonal transformations to input features at each time step via Patchwise Feature Diversification and Global Orthogonal Regularization. Experiments on CIFAR-10, CIFAR-100, and ImageNet show that STOD achieves better robustness against adversarial attacks than baselines.

**Strengths:**

1. The paper provides a mathematical analysis linking GTC to the spectral radius of the Hessian, connecting gradient structure directly to network robustness.

2. The proposed STOD framework improves robustness by decorrelating temporal features through orthogonal transformations. It delivers strong robustness gains on CIFAR-10, CIFAR-100, and ImageNet.

3. The paper offers clear Interpretability through visualization.

**Weaknesses:**

1. The link between the proposed GTC metric and the STOD method appears somewhat indirect. The paper does not clearly articulate how insights from GTC specifically motivate the design of STOD, nor does it provide sufficient theoretical or empirical evidence showing that STOD directly reduces GTC in a quantifiable manner.

2. The paper omits a detailed analysis of the computational overhead introduced by STOD during both training and inference. Without explicit comparisons of time or resource consumption against baseline methods, it is difficult to assess the practical efficiency and scalability of the proposed approach.

3. The method introduces several hyperparameters (e.g., patch size, regularization strength), yet the paper does not investigate whether the optimal settings generalize across different architectures and datasets. This raises concerns about the generalizability of the approach in broader applications.

4. The evaluation focuses primarily on static image datasets. However, SNNs are naturally well-suited for event-based or sequential data. The paper does not explore or discuss how STOD could be applied or adapted to event-based or sequential datasets, which limits the scope and generalizability of the proposed method.

**Questions:**

1. The robustness improvements of STOD appear smaller on ImageNet compared to CIFAR datasets. Could the authors explain the underlying reasons for this difference?

2. The experiments on black-box attacks and rate gradient approximation attacks did not include baseline comparisons. Could the authors clarify why these comparisons were omitted and whether STOD maintains its advantages when such baselines are included?

3. What are the key differences between the proposed STOD and the baseline methods? While it is intuitive that STOD improves over vanilla SNNs, it remains unclear why it also outperforms other robust approaches.

4. Could the authors empirically compare the robustness of direct coding with that of rate coding? Is the proposed STOD method still effective when applied to rate coding or other spike encoding schemes?

5. In Figure 1, GTC gradually decreases under direct coding. What reasons contribute to this trend?

6. How exactly is the gradient term G[i] in Eq. (3) computed? Does it aggregate gradients from all network parameters, or only from specific layers or subsets of weights?

7. Why is improving robustness particularly crucial for SNNs compared to ANNs? The authors might consider emphasizing the significance and potential benefits of robustness for SNNs in the Introduction.

8. Could the authors compare STOD with simpler baselines that inject random noise into features or gradients?

9. Why are adversarial attacks not applied at every time step? Would the proposed method remain effective under this more challenging attack setting?

---

> ### Author Response · Authors · 2025-11-28
> **Response to Reviewer 6hhX, part 1/7**
>
> ### **Weakness 1**:
>
> The link between the proposed GTC metric and the STOD method appears somewhat indirect. The paper does not clearly articulate how insights from GTC specifically motivate the design of STOD, nor does it provide sufficient theoretical or empirical evidence showing that STOD directly reduces GTC in a quantifiable manner.
>
> ### **Response:**
>
>  We thank the reviewer for this insightful comment. We clarify that the connection between GTC and the design of STOD is direct and intentional, and our revised manuscript (Sec. 4.2.1, highlighted in blue) further strengthens this link:
>
> The core idea of our method is to break the GTC, encouraging directional diversity across time steps so that external perturbations are misaligned in gradient space and their cumulative effect is mitigated. Since Sec. 4.1 shows that the high GTC of direct encoding originates from repeatedly injecting identical inputs across time while the low GTC of rate encoding arises from its inherent temporal diversity, a natural way to reduce GTC is to introduce controlled diversity into the temporal inputs of direct encoding. A seemingly intuitive approach to achieve this would be to inject random noise into the inputs or gradients to disrupt this repetition, but such noise lacks mechanism awareness and offers no guarantee of producing meaningful temporal diversity; moreover, these artificial perturbations do not correspond to the actual variations encountered by the network and may instead cause gradient obfuscation and reduce explainability. To achieve this reduction in an explainable and stable manner, we first require the temporal inputs to exhibit well-controlled and systematically varied feature directions, rather than random or unstructured perturbations, so that the resulting gradient components naturally become less aligned across time. Building on this requirement, we introduce a structured input transformation module, Patchwise Feature Diversification (PFD), which applies, at each time step, an individual parametric orthogonal kernel to transform the input features.
>
> Besides, regarding the evidence of GTC reduction, **we have already provided quantifiable evidence** that STOD directly lowers GTC. As shown in Fig. 6(c) and 6(d), increasing the strength of our global orthogonal regularization (\lambda_O) progressively decreases the epoch-averaged GTC throughout training. This demonstrates that STOD explicitly modifies the gradient temporal structure, validating both the motivation and the intended effect of the method. To our knowledge, this is the first SNN defense approach to empirically show a direct reduction of temporal gradient collinearity.
>
>
> ### **Weakness 2**:
>
> The paper omits a detailed analysis of the computational overhead introduced by STOD during both training and inference. Without explicit comparisons of time or resource consumption against baseline methods, it is difficult to assess the practical efficiency and scalability of the proposed approach.
>
> ### **Response**
>
> We thank the reviewer for raising this point. The computational overhead of STOD is in fact very limited, and we have highlighted this in the revised manuscript (Appendix F).
>
> During training, the only additional components are the orthogonal kernels. Because they operate on patches of size ($C \times p \times p$), the number of extra parameters is modest. As detailed in Appendix F, under our main configuration ($p = 8$), this results in only **0.15M additional parameters**, which is negligible compared to the size of VGG-11 or NF-ResNet-18. Thus, the training-time computational increase is minimal.
>
> During inference, we provide two modes:
>
> 1. **STOD w.o. OK (default mode)**: the orthogonal kernels are entirely removed, resulting in **zero extra inference cost**; runtime and memory usage are identical to the vanilla SNN baseline.
>
> 2. **STOD with OK (optional enhanced mode)**: keeping the kernels yields an additional **0.15M parameters** at inference, still very small relative to the backbone, and this mode further improves robustness, as shown in Table 2.
>
> Therefore, STOD’s inference-time overhead is either **exactly zero** or **extremely small**, depending on whether the user chooses the enhanced mode. The training overhead is also lightweight due to the compact patchwise design. We have highlighted these points in the revised manuscript for better readability.

---

> ### Author Response · Authors · 2025-11-28
> **Response to Reviewer 6hhX, part 2/7**
>
> ### **Weakness 3**:
>
> The method introduces several hyperparameters (e.g., patch size, regularization strength), yet the paper does not investigate whether the optimal settings generalize across different architectures and datasets. This raises concerns about the generalizability of the approach in broader applications.
>
> ### **Response**:
>
> We thank the reviewer for this observation. Although STOD introduces two hyperparameters, our ablation studies show that the method is **not brittle** with respect to their values. As we now emphasize in the revised manuscript (highlighted in blue), both patch size p and regularization strength exhibit smooth performance curves: suboptimal settings still provide substantial robustness gains over the vanilla SNN, and no abrupt performance degradation occurs when either hyperparameter is moderately mis-tuned. This indicates that STOD operates within a broad viable window rather than relying on finely tuned hyperparameters. Furthermore, the same hyperparameter values are used across CIFAR-10, CIFAR-100, and ImageNet, demonstrating that they generalize reliably across datasets and architectures. We have added a clarifying paragraph in Sec. 5.2 to highlight this non-brittle behavior as follows:
>
> Beyond identifying the optimal values of each hyperparameter, the ablation results collectively indicate that STOD exhibits stable behavior rather than brittle sensitivity across a broad range of settings. For patch size $p$, performance varies smoothly as $p$ changes; suboptimal values (e.g., $p=4$ or $p=16$) still yield substantial robustness gains over the vanilla SNN, showing that STOD does not collapse when deviating from its optimum. A similar pattern emerges for the regularization strength $\lambda_{\mathcal{O}}$: although $\lambda_{\mathcal{O}} = 0.05$ performs best, all non-zero values consistently reduce GTC and improve robustness compared with the baseline, and no abrupt degradation is observed even when the regularization is moderately mis-tuned. This robustness plateau demonstrates that STOD benefits from a wide viable hyperparameter window, enabling strong generalization across datasets and architectures without precise tuning.
>
> ### **Weakness 4**:
>
> The evaluation focuses primarily on static image datasets. However, SNNs are naturally well-suited for event-based or sequential data. The paper does not explore or discuss how STOD could be applied or adapted to event-based or sequential datasets, which limits the scope and generalizability of the proposed method.
>
> ### **Response**
>
> Thank you for the insightful comment. To address this concern, we have extended STOD to event-based datasets and conducted additional experiments on DVS-CIFAR10 (48×48 resolution) and DVS-Gesture (128×128 resolution). Notably, STOD can be applied to both datasets without any modification to the hyperparameters used for static-image experiments, demonstrating that the method transfers naturally to sequential and event-driven inputs. The results, as follows, show that STOD also improves robustness on both DVS datasets.
>
> |        | DVS-CIFAR10 |        |        | DVS-Gesture |        |        |
> |--------|-------------|--------|--------|-------------|--------|--------|
> | method | Clean       | FGSM   | PGD    | Clean       | FGSM   | PGD    |
> | SNN    | 76.30       | 17.20  | 5.00   | 95.49       | 39.24  | 9.72   |
> | SR     | 75.50       | 64.60  | 61.20  | 93.75       | 87.15  | 54.67  |
> | STOD w.o. OK | 74.70 | 66.10  | 60.40  | 93.05       | 88.88  | 56.60  |
> | STOD   | 72.90       | 68.80  | 62.50  | 93.05       | 88.88  | 56.60  |
>
> We have incorporated these experimental results and the above discussion into Sec. 5.1 of the revised manuscript and highlighted the additions in blue. Additionally, the introductions of DVS datasets and hyperparameter settings for DVS experiments are also included and highlighted in the Appendix.

---

> ### Author Response · Authors · 2025-11-28
> **Response to Reviewer 6hhX, part 3/7**
>
> ### **Question 1**:
>
>  The robustness improvements of STOD appear smaller on ImageNet compared to CIFAR datasets. Could the authors explain the underlying reasons for this difference?
>
> ### **Response**:
>
> Thank you for the insightful question. The smaller robustness gains on ImageNet primarily stem from the structural properties of high-resolution images and the deeper backbone used in this setting. ImageNet inputs contain much richer spatial information, and deeper architectures naturally diversify intermediate representations across layers, which reduces the temporal redundancy that STOD is designed to address. Consequently, the baseline SNN already exhibits lower gradient temporal collinearity on ImageNet than on CIFAR datasets, leaving a narrower margin for improvement. We have added a brief explanation in the revised manuscript (highlighted in blue).
>
> While the robustness gains on ImageNet are smaller than those on CIFAR datasets, this is expected. ImageNet’s higher-resolution inputs and the deeper backbone naturally produce richer and more diverse representations, yielding lower temporal redundancy. As a result, STOD has a narrower margin for improvement, though it still provides consistent robustness gains under this challenging setting.
>
> ### **Question 2**:
>
> The experiments on black-box attacks and rate gradient approximation attacks did not include baseline comparisons. Could the authors clarify why these comparisons were omitted and whether STOD maintains its advantages when such baselines are included?
>
> ### **Response**:
>
> Thank you for raising this point about black box attack and RGA attack. We intentionally refrain from comparing our method with other defenses under black-box settings because such evaluations do not reliably reflect true robustness. Even when all methods share the same surrogate model, the effectiveness of black-box attacks is largely governed by transferability, which itself depends on model-specific factors such as architectural biases, gradient structure, and optimization dynamics. Consequently, the observed attack success rates primarily capture how similar the target model is to the surrogate, rather than the genuine robustness of the defense. This makes black-box comparisons inappropriate for establishing a fair ranking across methods.
>
> In our work, black-box evaluation serves two focused purposes:
>
> * to verify that STOD remains effective under transfer attacks, and
>
> * to confirm that STOD does not exhibit gradient obfuscation, since a higher black-box accuracy relative to white-box accuracy aligns with the diagnostic criteria summarized in Table 4.
>
> Following the reviewer’s suggestion, we additionally performed black-box comparisons against several representative SOTA defenses. Because the validity of these comparisons is inherently constrained by surrogate–target mismatch, we did not integrate these results into the main manuscript. Instead, we present them in this rebuttal for completeness, and the corresponding experimental setup and outcomes are provided below.
>
> |        | CIFAR-10 |        |        | CIFAR-100 |        |        |
> |--------|----------|--------|--------|-----------|--------|--------|
> | Method | Clean    | FGSM   | PGD    | Clean     | FGSM   | PGD    |
> | SNN    | 93.75    | 10.26  | 0.89   | 72.39     | 9.16   | 0.78   |
> | AT     | 91.16    | 54.35  | 28.12  | 69.69     | 23.15  | 11.57  |
> | HoSNN  | 90.00    | 63.98  | 42.63  | 64.64     | 26.97  | 16.66  |
> | FEEL   | 92.70    | 61.20  | 56.70  | 74.20     | 28.80  | 19.40  |
> | STOD   | 90.87    | 75.42  | 44.24  | 70.69     | 41.96  | 25.90  |
>
> The experimental results of the performance of our method under black box attack can be found in our Appendix, specifically in Appendix E table 9. The above experimental results clearly show that our method is only slightly inferior to FEEL under the PGD attack of CIFAR-10. In other cases, **our method can achieve best robustness**.
>
> As for RGA, which is a specialized attack tailored specifically for SNNs, and its use in the existing literature remains relatively limited. To the best of our knowledge, there is no clearly established set of SOTA defenses that have been systematically benchmarked under RGA.  As a result, we were unable to identify comparable results from recent SNN defense studies that could serve as meaningful baselines for a fair comparison. If the reviewer is aware of any works that report standardized RGA evaluations on SOTA SNN defenses, we would be grateful for the reference and are more than willing to incorporate such comparisons.

---

> ### Author Response · Authors · 2025-11-28
> **Response to Reviewer 6hhX, part 4/7**
>
> ### **Question 3**:
>
> What are the key differences between the proposed STOD and the baseline methods? While it is intuitive that STOD improves over vanilla SNNs, it remains unclear why it also outperforms other robust approaches.
>
> ### **Response**:
>
> Thanks for your interest. The key difference between STOD and existing robust SNN approaches lies in the mechanism it targets. Most prior defenses operate on spatial representations, architectural modifications, or noise-based regularization, but none of them explicitly address the temporal gradient alignment that we identify through the GTC analysis. Our method is designed precisely to reduce this temporal collinearity, which is a robustness bottleneck intrinsic to direct encoding and not handled by previous approaches.
>
> Because STOD introduces controlled temporal feature diversification, rather than relying on stochastic perturbations or architectural heuristics, it fundamentally alters the gradient dynamics across time steps. This mechanism is complementary to existing defenses, and not a variation of them, which explains why STOD yields consistent improvements even over strong baselines. As shown in our experiments, the gains appear across multiple datasets, architectures, and attack types.
>
> ### **Question 4**:
>
> Could the authors empirically compare the robustness of direct coding with that of rate coding? Is the proposed STOD method still effective when applied to rate coding or other spike encoding schemes?
>
> ### **Response**:
>
> Thank you for the question. STOD is specifically designed as a robustness enhancement mechanism for direct encoding under BPTT, which is the mainstream setting for efficient SNN training. As analyzed in Sec. 4.1, the robustness weakness of direct encoding originates from its repeated injection of identical inputs, which leads to high gradient temporal collinearity. STOD directly targets this mechanism by introducing controlled temporal feature diversification. Because this issue is specific to direct encoding, STOD **is not** intended as a universal defense across all spike-encoding schemes. Applying STOD to other encoding strategies, such as Poisson rate coding, is possible, but the expected effect is inherently limited. Rate encoding naturally introduces temporal diversity through stochastic spike sampling, which already suppresses gradient alignment. Therefore, the margin for further improvement is smaller.
>
> Nonetheless, to address the reviewer’s suggestion, we conducted a supplementary experiment comparing STOD against Poisson rate encoding ($T$ = 32). The results, as shown below, demonstrate that **STOD still improves robustness** under rate encoding, but the gains are notably smaller than those observed in the direct-encoding setting. This outcome aligns with our theoretical analysis and also demonstrates that STOD remains compatible with alternative encoding schemes, even though they are not its primary design target.
>
> |        | Cifar-10 |        |        | cifar-100 |        |        |
> |--------|---|----|----|----|-----|----|
> | method | Clean    | FGSM   | PGD    | Clean     | FGSM   | PGD    |
> | Direct | 93.75    | 8.19   | 0.03   | 72.39     | 4.55   | 0.19   |
> | Direct+STOD | 91.43 | 55.80 | 32.97 | 71.00 | 26.26 | 13.13 |
> | Rate   | 82.31    | 20.34  | 3.16   | 61.01     | 11.89  | 2.16   |
> | Rate+STOD | 79.52 | 30.52 | 7.65 | 58.67 | 16.08 | 4.44 |
>
> ### **Question 5**:
>
> In Figure 1, GTC gradually decreases under direct coding. What reasons contribute to this trend?
>
> ### **Response**:
>
> Thank you for the question. As discussed in the paper, we believe the gradual decrease of GTC under direct encoding reflects a general property of the optimization dynamics rather than an effect tied to the time-step setting. During training, gradient updates progressively reshape the parameter space, causing the gradient components across time steps to become less aligned. In other words, the optimization trajectory naturally disperses gradient directions, reducing GTC even without any explicit decorrelation mechanism. This is consistent with our additional designed experiment using a trainable noise kernel (See Response to Reviewer ZQzg, weakness3, for details), where we observe the same phenomenon: the learned noise injection in the input feature gradually diversifies over training.
>
> The underlying reason is that, as learning progresses, the network increasingly allocates different functional roles to different time steps, e.g., early steps capturing coarse information and later steps refining decisions, which inherently reduces temporal redundancy and lowers the directional similarity of gradients. While this trend has not been explored in prior SNN literature, it suggests that GTC reduction may be an intrinsic emergent behavior of BPTT-trained SNNs. We believe this observation offers an interesting direction for future work, as it may reveal deeper principles about how temporal structure and learning dynamics interact in SNNs.

---

> ### Author Response · Authors · 2025-11-28
> **Response to Reviewer 6hhX, part 5/7**
>
> ### **Question 6**:
>
>  How exactly is the gradient term G[i] in Eq. (3) computed? Does it aggregate gradients from all network parameters, or only from specific layers or subsets of weights?
>
> ### **Response**:
>
> We thank the reviewer for the question. In Eq. (3), the gradient term $G[i]$ is computed from the **full set of trainable parameters** and corresponds to the gradient of the instantaneous loss at time step i. During SNN training, we accumulate the loss at each time step as:
>
> ```
> losses = []
> for t in range(time_step):
>     loss_t = ...
> losses.append(loss_t)
> ```
>
>
> For each time step t, we compute the gradient of loss_t with respect to all trainable parameters in the model:
>
> ```
> grads_t = torch.autograd.grad(
>     loss_t,
>     model.parameters(),
>     retain_graph=True,
>     allow_unused=True
> )
> ```
>
> Each gradient tensor is then flattened and concatenated into a single vector:
>
> ```
> def _flatten_and_concat(grads):
>     pieces = [g.reshape(-1).contiguous() for g in grads if g is not None]
> return torch.cat(pieces, dim=0)
> ```
>
> This gives the full-model gradient vector
>
> $G[i] = concat(flatten(\partial{loss_i}/\partial {\theta_1}), flatten(\partial{loss_j}/\partial {\theta_2}), … )$
>
> Where $\theta_1$, $\theta_2$, … denote all trainable parameters across the entire network. No layer-wise aggregation or partial selection is applied.
>
>
> ### **Question 7**:
>
> Why is improving robustness particularly crucial for SNNs compared to ANNs? The authors might consider emphasizing the significance and potential benefits of robustness for SNNs in the Introduction.
>
> ### **Response**:
>
> We thank the reviewer for this helpful suggestion, which makes our introduction more logical and appealing. In the revised manuscript, we have added a short paragraph in the Introduction (highlighted in blue) explaining why robustness is particularly crucial for SNNs. Details as follows:
>
> > *In addition to efficiency and high performance, robustness during real-world deployment is equally critical for SNNs. Due to their cumulative membrane-potential dynamics, small input perturbations can be repeatedly propagated and amplified over time, making SNNs inherently more vulnerable to temporal perturbation accumulation than traditional artificial neural networks. In many of the safety-critical domains where SNNs are most attractive, such as autonomous driving, robotics, and edge intelligence, this temporal sensitivity makes robustness not an optional enhancement but a fundamental requirement.*

---

> ### Author Response · Authors · 2025-11-28
> **Response to Reviewer 6hhX, part 6/7**
>
> ### **Question 8**:
>
> Could the authors compare STOD with simpler baselines that inject random noise into features or gradients?
>
> ### **Response**:
>
> This is entirely feasible. However, these baselines are ad-hoc and were constructed specifically for this rebuttal without undergoing systematic tuning or optimization; therefore, we do not consider them sufficiently stable or representative to be included as official baselines in the main manuscript. We hope the reviewer will understand this decision.
> We report the results in two parts.
>
> **1. Injecting trainable noise into the input features.**
>
> This experiment corresponds to our response to reviewer ZQzg’s Weakness 3, where we describe the design rationale, implementation details, and empirical results of this noise-based baseline. Due to its length, we kindly ask the reviewer to refer to that section. The key experimental results are summarized below:
>
> |        | CIFAR-10 |        |        | CIFAR-100 |        |        |
> |--------|----------|--------|--------|-----------|--------|--------|
> | Method | Clean    | FGSM   | PGD    | clean     | FGSM   | PGD    |
> | SNN    | **93.75**    | 8.19   | 0.03   | **72.39**     | 4.55   | 0.19   |
> | NKSNN w.o. NK | 92.49 | 11.01 | 0.49 | 71.99 | 7.43 | 0.72 |
> | NKSNN | 92.21 (1.21) | 13.87 (0.48) | 1.81 (0.13) | 71.83 (1.06) | 8.30 (0.35) | 0.91 (0.07) |
> | STOD w.o. OK | 91.43 | 55.80 | 32.97 | 71.00 | 26.26 | 13.13 |
> | STOD | 90.87 | **59.16** | **36.72** | 70.69 | **32.15** | **15.02** |
>
> Here, NKSNN refers to our temporary baseline “Noise Kernel SNN.” “NKSNN w.o. NK” denotes inference without the noise kernel, whereas “NKSNN” denotes inference with the noise kernel enabled. Because this setting introduces randomness at inference time, we report the mean accuracy over 10 inference runs along with the standard deviation in parentheses. As shown in the results, injecting noise into the input features indeed provides some robustness improvement, as one would intuitively expect, but the improvement remains limited and falls far short of what STOD achieves.
>
> **2. Injecting noise directly into gradients.**
>
> Our initial plan was to inject trainable noise into gradients, analogous to the noise-kernel design above. However, we found that this approach severely destabilized training, likely due to sensitivity to the learning-rate schedule. Despite repeated attempts, we were unable to obtain a usable configuration within the limited rebuttal timeframe, and we apologize for this. Consequently, in this section we report a simplified version in which manually controlled noise is added to the gradients directly. The corresponding code is shown below:
>
> ```
> def grad_noise(model, a=1e-7):
>     with torch.no_grad():
>         for param in model.parameters():
>             if param.grad is None:
>                 continue
>             g = param.grad.data
>             g_norm = g.norm()
>             noise = torch.randn_like(g)
>             n_norm = noise.norm()
>             noise = noise / n_norm * (a * g_norm)
>             param.grad.data.copy_(g + noise)
> ```
>
> ```
> loss.backward()
> grad_noise(model, a=args.noise)
> optimizer.step()
> ```

---

> ### Author Response · Authors · 2025-11-28
> **Response to Reviewer 6hhX, part 7/7**
>
> ### **Continue to response question 8**
>
> We trained the method under multi-scale of noise intensities on both CIFAR-10 and CIFAR-100 datasets, results as shown below (random seed=2025):
>
> |        | CIFAR-10 |        |        | CIFAR-100 |        |        |
> |--------|----------|--------|--------|-----------|--------|--------|
> | Method | Clean    | FGSM   | PGD    | Clean     | FGSM   | PGD    |
> | SNN    | **93.75**    | 8.19   | 0.03   | **72.39**     | 4.55   | 0.19   |
> | FNSNN a = 1e-7 | 85.13 | 11.14 | 0.09 | 61.57 | 6.72 | 0.10 |
> | FNSNN a = 5e-7 | 80.01 | 13.02 | 0.13 | 58.67 | 7.91 | 0.17 |
> | FNSNN a = 1e-6 | 82.86 | 11.98 | 0.11 | 53.34 | 9.87 | 0.26 |
> | FNSNN a = 5e-6 | 79.10 | 13.92 | 0.15 | 56.99 | 7.29 | 0.20 |
> | FNSNN a = 1e-5 | 75.16 | 14.81 | 0.14 | 48.93 | 11.87 | 0.24 |
> | FNSNN a = 5e-5 | 68.60 | 16.27 | 0.17 | 40.92 | 14.17 | 0.29 |
> | STOD w.o. OK | 91.43 | 55.80 | 32.97 | 71.00 | 26.26 | 13.13 |
> | STOD | 90.87 | **59.16** | **36.72** | 70.69 | **32.15** | **15.02** |
>
> It can be seen that when the random seed is set to 2025, the network becomes extremely sensitive to noise injected into the gradients. Even when the noise intensity is as small as 1e-7, the performance degradation remains substantial. Overall, these experiments show that directly injecting noise into gradients can provide a certain degree of robustness enhancement, but it also significantly harms clean accuracy, which is mainly due to the fact that the injected noise perturbs the optimization trajectory at every iteration, causing accumulated deviations in weight updates that amplify over training.
>
> On the other hand, gradient-noise injection is highly unstable. Under the same experimental configuration on CIFAR-10 but with a different random seed (seed = 42), the obtained results differ substantially from those under seed = 2025, see table as follows. This indicates that although the method can sometimes be effective, its training behavior is extremely unstable and highly dependent on random initialization. Due to limited time during the rebuttal period, we were unable to complete the seed = 42 experiment on CIFAR-100. However, based on the observed sensitivity on CIFAR-10, it is reasonable to expect that the behavior on CIFAR-100 would follow a similar pattern.
>
>
>
>
>
>
>
> | a | Seed | Clean | FGSM | PGD |
> |--------|------|--------|--------|--------|
> | -1e-7 | 2025 | 85.13 | 11.14 | 0.09 |
> |                | 42   | 87.35 | 13.72 | 0.10 |
> | -5e-7 | 2025 | 80.01 | 13.02 | 0.13 |
> |                | 42   | 79.88 | 12.21 | 0.20 |
> | -1e-6 | 2025| 82.86 | 11.98 | 0.11 |
> |                | 42   | 80.39 | 14.78 | 0.15 |
> | -5e-6 | 2025 | 79.10 | 13.92 | 0.15 |
> |                | 42   | 76.05 | 14.06 | 0.11 |
> | -1e-5 | 2025 | 75.16 | 14.81 | 0.14 |
> |                | 42   | 77.16 | 14.32 | 0.16 |
> | -5e-5 | 2025 | 68.60 | 16.27 | 0.17 |
> |                | 42   | 70.81 | 14.68 | 0.17 |
>
>
>
> ### **Question 9**:
>
> Why are adversarial attacks not applied at every time step? Would the proposed method remain effective under this more challenging attack setting?
>
> ### **Response**:
>
> Thank you for the question. We would like to clarify that in our implementation, **adversarial attacks are indeed applied at every time step**. For both FGSM and PGD, the perturbation is generated from the input sample x, and under direct encoding the same (clean or adversarial) input is injected into the SNN at each time step. Therefore, attacks are not applied at only a subset of time steps; they affect all time steps equally, exactly following the standard adversarial setting used in prior SNN studies.
>
> Regarding the reviewer’s question about whether STOD remains effective under this setting: yes, all robustness gains reported in the paper are obtained under this “worst-case” per-time-step attack scenario. Since perturbations are already injected at every time step by design, the evaluation setting is fully aligned with the more challenging attack condition mentioned by the reviewer.
>
> Specifically, you can find the following logic in our code provided in the supplementary materials, path: \STOD code\utils\tvc.py :
>
> ```
> if attacker is not None:
>     input_frame = attacker(frame, label)
>
> for t in range(time_step):
>     out = model(input_frame)

---

### Official Review · Reviewer_ZQzg · 2025-10-29

**Soundness:** 2
**Presentation:** 3
**Contribution:** 1
**Rating:** 2
**Confidence:** 4

**Summary:**

The authors study the resistance of spiking neural networks (SNNs) to adversarial attacks (white box and black box).

They argue that direct input coding, which outperforms Poisson rate coding in terms of accuracy, makes the SNNs less resistant because of strong correlations across timesteps.

They propose a new method to decorrelate timesteps, and thus increase resistance to adversarial attacks.

**Strengths:**

As far as I know, the method is new, and it outperforms other approaches for white box attacks (Table 1).

**Weaknesses:**

I am not convinced, because a number of comparisons are missing:

* Comparison to SOTA (Table 1) is limited to white box attacks. How about for black box attacks?

* The authors should also compare their method to Poisson rate coding.

* How about using direct input coding, but adding white noise? This should also decorrelate time steps (similar to Poisson rate coding). One could vary the noise amplitudes. Maybe there is a sweet spot in terms of amplitude, the authors should investigate.

* An alternative to direct input coding has been proposed in
https://dl.acm.org/doi/10.5555/3692070.3692856
the idea is simple:
a simple 2d conv is applied to the input image with:
Cin = 3 (RGB channels)
Cout = T
the output is [T, W, H] then at each t timestep t we they feed the image [t, :, :] to the SNN.
This could also decorrelate timesteps.
It would be interesting to compare the proposed approach to this previous one, in terms of accuracy and resistance to attacks.

**Questions:**

The size of the orthogonal kernels is d in the text, but c x p x p on Fig 1. So you choose d = c x p x p?

---

> ### Author Response · Authors · 2025-11-28
> **Response to Reviewer ZQzg, part 1/6**
>
> ### **Weakness 1**:
>
> Comparison to SOTA (Table 1) is limited to white box attacks. How about for black box attacks?
>
>
> ### **Response**:
>
> Thank you for the question. We deliberately avoid comparing different defense methods under black box attacks because such comparisons are not fair or interpretable, even when the same substitute model is used. Black box attack strength fundamentally depends on transferability, which varies across defense methods due to differences in their architectures, gradient geometry, and training dynamics. Thus, even with an identical substitute model, the attack success rate reflects model–model similarity, not the intrinsic robustness of each method. This makes black box results unsuitable for ranking defenses.
>
> For this reason, in our paper, we include black box experiments for two specific purposes:
>
> * **To show that STOD is also robust under black box transfer attacks**, demonstrating that its benefits are not limited to the whit box setting.
>
> * **To validate that STOD does not rely on gradient obfuscation**, as black box accuracy being higher than whit box accuracy exactly matches the second check criterion in Table 4.
>
> In response to the reviewer’s suggestion, we have additionally conducted a set of black box comparisons with representative SOTA methods. Because such comparisons depend heavily on surrogate–target transferability and thus do not provide fair methodological ranking, we have chosen not to place these results in the main manuscript. Instead, we report them here in the rebuttal for completeness. The experimental details and results are provided below.
>
> |        | CIFAR-10 |        |        | CIFAR-100 |        |        |
> |--------|--------|--------|--------|--------|--------|--------|
> | Method | Clean    | FGSM   | PGD    | Clean     | FGSM   | PGD    |
> | SNN    | 93.75    | 10.26  | 0.89   | 72.39     | 9.16   | 0.78   |
> | AT     | 91.16    | 54.35  | 28.12  | 69.69     | 23.15  | 11.57  |
> | HoSNN  | 90.00    | 63.98  | 42.63  | 64.64     | 26.97  | 16.66  |
> | FEEL   | 92.70    | 61.20  | 56.70  | 74.20     | 28.80  | 19.40  |
> | STOD   | 90.87    | 75.42  | 44.24  | 70.69     | 41.96  | 25.90  |
>
> The experimental results of the performance of our method under black box attack can be found in our Appendix, specifically in Appendix E table 9. The above experimental results clearly show that our method is only slightly inferior to FEEL under the PGD attack of CIFAR-10. In other cases, our method can achieve best robustness.
>
>
> ### **Weakness 2**:
>
> The authors should also compare their method to Poisson rate coding.
>
> ### **Response**:
>
> We agree that Poisson rate encoding is an important reference point, and indeed our paper already includes a detailed analytical comparison between direct encoding and rate encoding through GTC (Fig. 1), where rate encoding consistently exhibits lower GTC. This comparison was intentionally placed in the analysis section rather than the robustness benchmark tables, for the following reason.
>
> A fair empirical comparison between STOD (based on direct encoding) and Poisson rate encoding is not feasible because the robustness and accuracy of rate encoding are highly sensitive to the number of time steps $T$, as [1] investigated the robustness between rate encoding and direct encoding. However, the two encoding paradigms have fundamentally different constraints:
>
> 1. Direct encoding is designed to operate with very small $T$ (e.g., $T$ = 4) and cannot practically benefit from large $T$.
>
> 2. Rate encoding requires large $T$ (typically 32–128) to approximate firing statistics; reducing $T$ to match direct encoding leads to a severe and uninformative accuracy collapse.
> Therefore:
>
> •	If we force $T$ to be equal, rate encoding becomes an artificially weak baseline due to insufficient sampling.
>
> •	If we allow rate encoding to use its appropriate large $T$, then the comparison becomes confounded by dramatically different computational budgets, violating fairness in both robustness and efficiency evaluation.
>
> Nonetheless, to address the reviewer’s suggestion, we can include a supplementary experiment in the rebuttal comparing STOD to Poisson rate encoding ($T$ = 64).
>
> |        | Cifar-10 |        |        | cifar-100 |        |        |
> |--------|--------|--------|--------|--------|--------|--------|
> | method | Clean   | FGSM   | PGD    | Clean    | FGSM   | PGD    |
> | Direct | 93.75   | 8.19   | 0.03   | 72.39    | 4.55   | 0.19   |
> | Rate   | 85.08   | 22.58  | 4.51   | 63.12    | 13.17  | 3.75   |
> | STOD   | 91.43   | 55.80  | 32.97  | 71.00    | 26.26  | 13.13  |
>
> It can be clearly seen that our method is significantly superior to the SNN under Poisson rate encoding in both Clean accuracy and attack accuracy.
>
> > [1] Rate coding or direct coding: Which one is better for accurate, robust, and energy-efficient spiking neural networks?, ICASSP, 2022.

---

> ### Author Response · Authors · 2025-11-28
> **Response to Reviewer ZQzg, part 2/6**
>
> ### **Weakness 3:**
>
> How about using direct input coding, but adding white noise? This should also decorrelate time steps (similar to Poisson rate coding). One could vary the noise amplitudes. Maybe there is a sweet spot in terms of amplitude, the authors should investigate.
>
> ### **Response:**
>
> We greatly appreciate the reviewer’s interest in adding noise to the direct-encoding input. If identical noise is applied to every time step, it cannot meaningfully break temporal correlations. If different noise patterns are used across time steps, the noise must be learned through optimization rather than manually injected, as manually fixed noise is not compatible with training dynamics and is unlikely to yield favorable results. Prior work such as that of [1]. has also introduced input noise during training, but their approach was not designed from the perspective of temporal feature decorrelation and does not include such analysis. For this discussion, we therefore implemented a minimal trainable noise kernel to explore feasibility and provide a brief analysis from the perspective of temporal decorrelation.
>
> There are two parts: 1. Method design and, 2. Result and analysis, specifically:
>
> ### **1. Method design**
>
> In terms of the overall idea of the method, we create a trainable noise kernel before the input layer to add noise to the input data:
>
> **1.1 Noise Kernel Definition**
>
> Given an SNN with $T$ time steps, $C$ input channels, and input frames $x_t \in \mathbb{R}^{B \times C \times H \times W}$,
>
> we define a trainable noise kernel
>
> $K \in \mathbb{R}^{T \times C},$
>
> where each element $K_{t,c}$ controls the noise strength for time step $t$ and channel $c$.
>
> To ensure stable optimization, the effective noise magnitude is bounded through a sigmoid parameterization:
>
> $\alpha_{t,c} = \varepsilon_{\max} \times \sigma(K_{t,c}),$
>
> where $\varepsilon_{\max}$ is the maximum allowable noise scale.
>
> This produces a temporal–channel noise profile
>
> $\alpha_t = (\alpha_{t,1},\ldots,\alpha_{t,C}) \in \mathbb{R}^C, $
>
> shared across all spatial locations.
>
> **1.2. Noise Injection**
>
> At each time step $t$, we sample Gaussian white noise
>
> $z_t \sim \mathcal{N}(0,1)^{B \times C \times H \times W},$
>
> and inject it into the normalized input:
>
> $\tilde{x}_t = x_t + \alpha_t \odot z_t,$
>
> where broadcasting is applied over $H, W$.
>
> Noise is added after all preprocessing and normalization, ensuring that perturbations are aligned with the actual input scale received by the SNN.
>
> **1.3. Initialization**
>
> We initialize all entries of $K$ to the same constant $K_0$, corresponding to a small initial noise level $\varepsilon_{\text{init}}$.
>
> From the parameterization:
>
> $\sigma(K_0) = \frac{\varepsilon_{\text{init}}}{\varepsilon_{\max}},$
>
> so the initial value is:
>
> $K_0 = \log \left(\frac{\varepsilon_{\text{init}}}{\varepsilon_{\max} - \varepsilon_{\text{init}}}\right).$
>
> Thus all time steps and channels begin with an identical noise scale:
>
> $ \alpha_{t,c} \approx \varepsilon_{\text{init}}, \quad \forall t,c.$
>
> This ensures stable early training and no artificially imposed temporal bias.

---

> ### Author Response · Authors · 2025-11-28
> **Response to Reviewer ZQzg, part 3/6**
>
> ### **Continue to response weakness 3**
>
> **1.4. Training and Evaluation Behavior**
>
> •	Training:
> The noise kernel is optimized jointly with the SNN. Gaussian noise is sampled independently at each time step.
>
> •	Evaluation:
> Noise kernel is optionally disabled for inference. If we choose to retain the noise kernel in the inference, then we use the multiple-time average for the inference result and retain the standard deviation.
>
> **1.5 Code implementation**
>
> PyTorch code implementation as follows:
>
> First, we define the noise kernel:
>
> ```
> class noise_kernel(nn.Module):
>     def __init__(self, time_step, channels, eps_max=0.3, eps_init=0.05):
>         super().__init__()
>         self.time_step = time_step
>         self.channels = channels
>         self.eps_max = eps_max
>
>         #initialization
>         p0 = eps_init / eps_max
>         a0 = math.log(p0 / (1.0 - p0))
>         a = torch.full((time_step, channels), a0)
>         self.a = nn.Parameter(a)
>
>     def forward(self, x, t):
>         alpha_tc = self.eps_max * torch.sigmoid(self.a[t])
>         alpha_tc = alpha_tc.view(1, self.channels, 1, 1)
>
>         # Gaussian white noise
>         noise = torch.randn_like(x)
>         return x + alpha_tc * noise
> ```
>
> Then we add this noise kernel to the model:
>
> ```
> model.noise = noise_kernel()
> optimizer = torch.optim.SGD(list(model.parameters()) + list(net.noise.parameters()))
> ```
>
> Finally, we add noise before the forward pass:
>
> ```
> input = model.noise(input_frame, t)
> ```
>
> ### **2. Result and analysis**
>
> We trained the SNN based on noise kernels (NKSNN) on CIFAR-10 and CIFAR-100 using clean data, and all hyperparameters are consistent with our method STOD. Here we list the performance comparisons among vanilla SNN, NKSNN and our method STOD. Here, NKSNN w.o. NK represents inference without using noise kernel, and NKSNN represents inference with noise kernel. Since there is randomness in the inference when using noise kernel, we use different random seeds for inference 10 times, take the average value, and list the standard deviations in parentheses.
>
> |        | CIFAR-10 |        |        | CIFAR-100 |        |        |
> |--------|--------|--------|--------|--------|--------|--------|
> | Method | Clean   | FGSM   | PGD    | Clean   | FGSM   | PGD    |
> | SNN    | 93.75   | 8.19    | 0.03          | 72.39   | 4.55    | 0.19          |
> | NKSNN w.o. NK | 92.49   | 11.01   | 0.49          | 71.99   | 7.43    | 0.72          |
> | NKSNN  | 92.21 (1.21) | 13.87 (0.48) | 1.81 (0.13) | 71.83 (1.06) | 8.30 (0.35) | 0.91 (0.07) |
> | STOD w.o. OK | 91.43   | 55.80   | 32.97         | 71.00   | 26.26   | 13.13         |
> | STOD   | 90.87   | 59.16   | 36.72         | 70.69   | 32.15   | 15.02         |
>
> From the experimental results, we observe that the noise-kernel approach indeed provides a moderate improvement in robustness while causing only a small drop in clean accuracy. However, its overall performance still remains noticeably below that of our proposed STOD. Moreover, comparing inference with the noise kernel versus without it shows that the presence or absence of the kernel has almost no effect on performance. This indicates that the function of the noise kernel—similar to the role of the orthogonal kernels in our method—is already **internalized into the learned network weights** during training, rather than depending on the noise injection at inference time. In other words, the kernel primarily guides the network to learn differentiated representations across time steps, rather than acting through its literal noise at test time.

---

> ### Author Response · Authors · 2025-11-28
> **Response to Reviewer ZQzg, part 4/6**
>
> ### **Continue to response weakness 3**
>
> In addition, we list the learned noise-kernel parameters under the settings $T$ = 4 and $T$ = 8. The first row corresponds to the three channel-wise parameters at the first time step, the second row corresponds to the second time step, and so on. These values allow us to analyze how the learned noise magnitudes evolve structurally across time.
>
> |                | C1       | C2       | C3       |
> |----------------|----------|----------|----------|
> | CIFAR-10 $T$=4   | -1.6825  | -1.5509  | -1.5357  |
> |                | -1.5228  | -1.3794  | -1.3709  |
> |                | -1.3822  | -1.2315  | -1.2208  |
> |                | -1.1119  | -0.9781  | -0.9697  |
> | CIFAR-100 $T$=4  | -1.6825  | -1.5509  | -1.5357  |
> |                | -1.5228  | -1.3794  | -1.3709  |
> |                | -1.3822  | -1.2315  | -1.2208  |
> |                | -1.1119  | -0.9781  | -0.9697  |
> | CIFAR-10 $T$=8   | -0.9786  | -0.9687  | -0.9697  |
> |                | -0.9480  | -0.9483  | -0.8886  |
> |                | -0.8930  | -0.8720  | -0.8188  |
> |                | -0.8282  | -0.8228  | -0.7623  |
> |                | -0.7803  | -0.7413  | -0.6743  |
> |                | -0.6562  | -0.6500  | -0.6028  |
> |                | -0.5354  | -0.5376  | -0.5023  |
> |                | -0.4159  | -0.4222  | -0.3947  |
> | CIFAR-100 $T$=8  | -1.4332  | -1.1813  | -1.1816  |
> |                | -1.3736  | -1.1098  | -1.1718  |
> |                | -1.3186  | -1.0879  | -1.0875  |
> |                | -1.2592  | -1.0144  | -1.0003  |
> |                | -1.1729  | -0.9316  | -0.9422  |
> |                | -1.0809  | -0.8299  | -0.8174  |
> |                | -0.8890  | -0.6636  | -0.6902  |
> |                | -0.6332  | -0.4588  | -0.4607  |
>
> Based on these trained parameters, we observe that the differences across channels within the same time step are very small. Therefore, we average the three channel-wise values at each time step and apply a sigmoid function, yielding the mean noise intensity for each time step as follows:
>
> |  | Timestep | CIFAR-10 | CIFAR-100 |
> |---|----------|----------|-----------|
> | $T$=4 | 1 | 0.1696 | 0.2246 |
> |  | 2 | 0.1940 | 0.2465 |
> |  | 3 | 0.2187 | 0.2676 |
> |  | 4 | 0.2650 | 0.3148 |
> | $T$=8 | 1 | 0.2743 | 0.2207 |
> |  | 2 | 0.2832 | 0.2324 |
> |  | 3 | 0.2971 | 0.2385 |
> |  | 4 | 0.3091 | 0.2520 |
> |  | 5 | 0.3280 | 0.2664 |
> |  | 6 | 0.3459 | 0.2893 |
> |  | 7 | 0.3717 | 0.3223 |
> |  | 8 | 0.3984 | 0.3739 |
>
> By inspecting the learned noise parameters, we observe that across time steps, the amplitudes exhibit a **clear monotonic increase**. This pattern appears consistently on both CIFAR-10 and CIFAR-100, $T$=4 and $T$=8, demonstrating that it is not randomness but a learned temporal behavior.
>
> These observations highlight three points:
>
> 1.	Injecting noise into the input can indeed enhance robustness, though with limited performance, and remains a promising future direction
>
> 2.	The noise amplitudes diverge across time steps, showing that the network naturally learns to break temporal consistency, consistent with our analysis of GTC in the paper.
>
> 3.	The monotonic increase of noise strength across time indicates that the network prefers to preserve more original information in earlier steps and introduce stronger perturbations in later steps, which aligns with the design of our first structural constraint in STOD, where the first orthogonal kernel is initialized as an identity mapping.
>
> In summary, the reviewer’s intuition is correct: noise injection can decorrelate temporal features and improve robustness to some extent. This supplemental investigation, however, is independent of the main method proposed in our paper and is included solely to illustrate feasibility.

---

> ### Author Response · Authors · 2025-11-28
> **Response to Reviewer ZQzg, part 5/6**
>
> ### **Weakness 4**:
>
>  An alternative to direct input coding has been proposed in https://dl.acm.org/doi/10.5555/3692070.3692856 the idea is simple: a simple 2d conv is applied to the input image with: Cin = 3 (RGB channels) Cout = T the output is [T, W, H] then at each t timestep t we they feed the image [t, :, :] to the SNN. This could also decorrelate timesteps. It would be interesting to compare the proposed approach to this previous one, in terms of accuracy and resistance to attacks.
>
> ### **Response**:
>
> We thank the reviewer for pointing out the reference [1]. However, the description “Cin = 3 (RGB channels) Cout = T the output is [T, W, H] then at each t timestep t, and feed the image [t, :, :] to the SNN” does not match the actual architecture used in that work, either in the paper or in the released code.
>
> In the official implementation, the RGB image is first passed through a standard convolutional stem:
>
> ```
> class MS_PatchEmbed(nn.Module):
>     def __init__(self, channels, num_subnet):
>         super().__init__()
>         self.down1 = ScaledStdConv2d(3, channels, kernel_size=7, stride=2, padding=3)
>         self.down2 = ScaledStdConv2d(channels, channels, kernel_size=3, stride=2, padding=1)
>
>     def forward(self, x):
>         x = self.down1(x)
>         x = F.leaky_relu(x, 1.0)
>         x = self.down2(x)
>         return None, x
> ```
>
> Thus, an input of shape $[B, 3, H, W]$ is encoded into a high-dimensional feature map x of shape $[B, C_0, H′, W′]$, where $C_0$ is the first embed dimension (e.g., 32/64/96) determined by the model configuration. This is consistent with Table 7 in the appendix, where the authors explicitly list the embed dimensions as [64, 128, 256, 384] or [96, 192, 384, 512] for ImageNet. In particular, the number of channels $C_0$ **is much larger than** $T$; there is no place in the implementation where a convolution with Cout = $T$` is applied to the RGB input.
>
> The paper itself also states that “the image is encoded only once. The encoded features are divided into $T$ groups, exploited as input for each timestep. Correspondingly, the entire SNN is also divided into $T$ independent sub-networks…”. This means that $T$ arises from splitting the encoded features into $T$ groups along the channel dimension, not from using a convolution whose output channel dimension equals $T$. Concretely, they first obtain a feature tensor with $C_0$ channels, where $C_0$ is one of the large embed dimensions above, and then hard-partition these $C_0$ channels into $T$ groups (conceptually $C_0/T$ channels per group). Each group is then processed by a sub-network corresponding to one timestep. Therefore, it is not the case that the encoder directly produces a tensor of shape $[T, W, H]$ with only $T$ channels.
>
> This behavior is also reflected in the released model’s forward pass for static images:
>
> ```
> def _forward_intermediate_supervision(self, img):
>     _, x = self.stem(img)
>     for i in range(self.num_subnet): # num_subnet = T
>         c0, c1, c2, c3 = self.subnet_i(x.unsqueeze(0), c0, c1, c2, c3)
> ```
>
> Here, num_subnet (their $T$) controls how many sub-networks are applied, but there is no Conv2d(3, $T$, ·) and no tensor of shape $[T, W, H]$. Instead, the same high-dimensional encoded feature x is processed by a sequence of ConvNeXt-style SNN blocks (implemented in Level and SubNet) with multi-level temporal-reversible interactions, as described in their Sections 4.2–4.3.
>
> Nevertheless, we agree with the reviewer that the underlying intuition is meaningful—introducing additional input–level convolutions that decorrelate information across the temporal dimension may further enhance SNN behavior.
>
> We designed an additional method and conducted experiments to verify the reviewer’s intuition.
>
> First, we implemented a simple convolution-based temporal encoding layer at the input:
>
> ```
> self.time_conv = nn.Conv2d(
>     in_channels=3,
>     out_channels=time_step,
>     kernel_size=3,
>     padding=1
> )
> ```
>
> Under this configuration, each time step receives one slice from the $T$ output channels of the convolution, meaning that the first SNN layer effectively sees a single-channel input at each time step. We refer to this variant as **SNN+ConvEn_T**. However, because the original SNN receives a three-channel image at every time step, we anticipated that reducing the channel count to one may weaken feature representation. To address this issue, we further designed a variant:
>
> ```
> self.time_conv = nn.Conv2d(
>     in_channels=3,
>     out_channels=3 * time_step,
>     kernel_size=3,
>     padding=1
> )
> ```
> This ensures that **SNN+ConvEn_3T** feeds a full three-channel representation into the network at every time step, thus preserving the original feature capacity as much as possible.

---

> ### Author Response · Authors · 2025-11-28
> **Response to Reviewer ZQzg, part 6/6**
>
> ### **Continue to response weakness 4**
>
>
> We evaluated both variants on CIFAR-10 and CIFAR-100, and the corresponding robustness results are provided below:
>
> |        | CIFAR-10 |        |        | CIFAR-100 |        |        |
> |--------|----------|--------|--------|-----------|--------|--------|
> | Method | Clean    | FGSM   | PGD    | Clean     | FGSM   | PGD    |
> | SNN | 93.75 | 8.19 | 0.03 | 72.39 | 4.55 | 0.19 |
> | SNN+ConvEn_T | 87.10 | 12.43 | 0.14 | 60.61 | 7.17 | 0.29 |
> | SNN+ConvEn_3T | 88.90 | 15.57 | 0.30 | 64.97 | 9.12 | 0.48 |
> | STOD w.o. OK | 91.43 | 55.80 | 32.97 | 71.00 | 26.26 | 13.13 |
>
> As the reviewer anticipated, introducing a convolutional encoding layer indeed improves robustness, since the underlying mathematical effect, decorrelating features at the input layer, is similar in spirit to our STOD method. Moreover, the three-channel version (SNN+ConvEn_3T) consistently outperforms the single-channel variant on both clean and perturbed inputs, supporting the importance of preserving full feature richness.
>
> However, both variants suffer from a substantial drop in clean accuracy, likely due to the simplicity and lack of tuning in our initial design. At the same time, this observation highlights the potential of convolution-based temporal encoding as a promising direction. We plan to further explore this idea and develop a more principled formulation in future work, as it may become a valuable extension to robustness-oriented SNN training.
>
> > [1] High-performance temporal reversible spiking neural networks with O(L) training memory and O(1) inference cost, ICML, 2024.
>
>
> ### **Question 1**:
>
> The size of the orthogonal kernels is d in the text, but c x p x p on Fig 1. So you choose d = c x p x p?
>
> ### **Response**
>
> We thank the reviewer for the clarification request. Yes, the symbol d is used in the main text as a shorthand notation for the vectorized patch dimension
>
> $ d = C \times p \times p.$
>
> This is purely a notational simplification to make the orthogonal-kernel formulations more readable.
>
> Due to a writing oversight, this explicit relation was not stated in the submitted version. We apologize for the omission. In the revised manuscript, we have added this definition with highlighted in blue：
>
> >*… where each $O[t] \in \mathbb{R}^{dxd}$ with $d = C \times p^2$*

---

### Official Review · Reviewer_PFkG · 2025-10-31

**Soundness:** 3
**Presentation:** 3
**Contribution:** 3
**Rating:** 6
**Confidence:** 3

**Summary:**

This paper proposes Structured Temporal Orthogonal Decorrelation (STOD), a framework designed to mitigate Gradient Temporal Collinearity (GTC) - the tendency of gradients in Spiking Neural Networks to align their directions across time steps, where high GTC in direct encoding results in vulnerability to adversarial perturbations.
STOO combines two complementary modules: Patchwise Feature Diversification (PFD), which enforces local orthogonality in kernel transformations to preserve intra-step feature independence; and Global Orthogonal Regularization (GOR), which introduces a feature-level angular diversity loss — a cosine-based soft constraint (Eq. 8) — to penalize similarity between normalized transformed features across time steps. This combination jointly ensures orthogonal kernel dynamics and temporally diverse representations, leading to robust and decorrelated spatio-temporal learning.
Empirical evaluations on CIFAR-10/100 and ImageNet under multiple adversarial attack scenarios demonstrate consistent improvements without additional inference overhead.

**Strengths:**

- Clearly identifies GTC as a key cause of robustness degradation in SNNs and provides an intuitive explanation of its geometric implications, supported by experiments (in Fig.1).
- Integrates local (PFD) and global (GOR) components to address temporal alignment of gradients from complementary kernel- and feature-level perspectives, offering a coherent and theoretically consistent framework.
- Evaluates robustness under diverse attack settings (e.g., FGSM, PGD and both white- and black-box attacks) and across multiple datasets, presenting consistent gains over baselines.
- Effectively connects temporal gradient decorrelation to robustness improvement while maintaining no inference overhead due to removal of orthogonal kernels at test time.

**Weaknesses:**

- Some mathematical formulations are ambiguous: in Eq.(7), the index $j$ range in the Householder reflection is unclear ($j \in [1,T]$ vs. $j \in [1,d]$), limiting reproducibility and interpretability of the initialization process.
- Lacks ablation studies analyzing the independent contributions of PFD and GOR, making it difficult to assess which component drives the main performance gains in STOD.
- Although the motivation contrasts direct and rate encoding, there is no quantitative evidence that STOD-enhanced direct encoding exhibits rate-like decorrelation characteristics. Fig.4 includes comparisons, but those differences are visually small and not statistically analyzed.
- Experiments are limited to static image benchmarks, which aligns with the focus on input encoding but leaves open whether the proposed decorrelation mechanism generalizes to other domains of natural inputs (such as audio, time-series, etc.).

**Questions:**

1. Could you provide ablation results isolating the impact of PFD and GOR individually? It is difficult to evaluate their relative contributions without such analysis.
2. In Fig. 4, the comparison between direct encoding and STDO shows only marginal visual differences, and the transformed features via PFD across time steps also appear similar. Could you provide quantitative evidence or clearer visualizations demonstrating improved gradient diversity?
3. Eq.(7) defines the canonical basis $e_1, …, e_d$, yet the index range for $j$ ($[1,T]$ vs. $[1,d]$) remains ambiguous. How is this handled to ensure valid kernel initialization?
4. The inference phase introduces no additional overhead because learned orthogonal kernels are discarded. Can you clarify whether inference without these kernels still maintains the same level of temporal decorrelation?
5. The paper focuses exclusively on static natural-image datasets. While this choice aligns with the motivation around direct encoding, have you considered extending STOD to other natural input domains (e.g., sensor time-series, audio, etc.), where gradient temporal collinearity may also naturally arise?

---

> ### Author Response · Authors · 2025-11-28
> **Response to Reviewer PFkG, part 1/4**
>
> ### **Weakness 1 & Question 3**:
>
> Some mathematical formulations are ambiguous: in Eq.(7), the index range in the Householder reflection is unclear ($j \in [1,T] vs. j \in [1,d] $), limiting reproducibility and interpretability of the initialization process.
>
> ### **Response:**
>
> Thank you for raising this point. Our mathematical definition and initialization procedure are fully correct; the ambiguity arises simply because we omitted one clarifying sentence in the original text. In our method, $d$ denotes the dimension of the orthogonal kernel, and $T$ is the number of time steps. Since in all experiments the patch dimension satisfies ($d = C \times p^2 \gg T$), only the first $T$ canonical basis vectors (${e_1,\ldots,e_T} \subset \mathbb{R}^d$) are needed to construct the reflection vectors ($k_j = e_1 - e_j$). Under this condition, the index range is well-defined and mathematically consistent, and the initialization remains entirely reproducible. We have added an explicit explanation as follows in the revised manuscript, which is highlighted in blue, to avoid any potential ambiguity for readers.
>
> > *We use the first $T$ canonical basis vectors since $T \le d$ in all settings.*
>
> ### **Weakness 2 & Question 1**:
>
> Lacks ablation studies analyzing the independent contributions of PFD and GOR, making it difficult to assess which component drives the main performance gains in STOD.
>
> ### **Response:**
>
> We appreciate the reviewer’s concern regarding the independent contributions of PFD and GOR. We clarify that in the STOD framework, GOR is not an independent module but the fourth structured soft constraint applied on top of PFD. In other words, PFD forms the core transformation mechanism, while GOR refines it by encouraging additional directional diversity across time steps. As a result, GOR cannot function without PFD, and isolating GOR as a standalone component is not structurally meaningful.
>
> For this reason, our ablation study evaluates GOR by varying its regularization strength $\lambda_{O}$. Setting $\lambda_{O}$ = 0 corresponds exactly to using PFD alone, with GOR fully removed. Increasing $\lambda_{O}$ gradually introduces the effect of GOR. This design provides a clean and valid way to measure the independent influence of GOR within the only feasible configuration of the STOD framework.
>
> The results in Fig. 6 (and Appendix E) show the following:
>
> * When $\lambda_{O}$ = 0, the model retains significant robustness improvements, indicating that PFD is the primary contributor.
>
> * Moderate $\lambda_{O}$ (e.g., 0.05) further reduces GTC and improves robustness, demonstrating the auxiliary benefit of GOR.
>
> * Excessive $\lambda_{O}$ (e.g., 0.1) harms performance, confirming that GOR should be viewed as a soft enhancement rather than a standalone module.

---

> ### Author Response · Authors · 2025-11-28
> **Response to Reviewer PFkG, part 2/4**
>
> ### **Weakness 3 & Question 2**:
>
> Although the motivation contrasts direct and rate encoding, there is no quantitative evidence that STOD-enhanced direct encoding exhibits rate-like decorrelation characteristics. Fig.4 includes comparisons, but those differences are visually small and not statistically analyzed.
>
> ### **Response**:
>
> We thank the reviewer for raising this point. Our response is organized as follows.
>
> 1.	**We have already provided quantitative evidence of decorrelation in the paper.** The primary quantitative indicator of temporal decorrelation in SNNs is the GTC. As shown in ablation study, Fig. 6 (and Appendix E), STOD consistently reduces the epoch-averaged GTC throughout training. Since GTC directly measures the alignment of gradient directions across time steps, its systematic reduction constitutes quantitative evidence that STOD introduces temporal decorrelation within direct encoding. This reduction also closely matches the robustness improvements, in line with the theoretical analysis in Sec. 4.1 and Appendix B.
>
> 2.	**STOD is not intended to exactly reproduce the GTC behavior of rate encoding.** Our design borrows the idea of temporal decorrelation from rate encoding, but does not incorporate its intrinsic randomness. To avoid any risk of gradient obfuscation [1], all components of STOD are fully deterministic, from initialization to training. Combined with the fact that direct encoding uses only a few time steps and repeatedly injects the same input, it is mathematically impossible for a deterministic transformation to achieve the same extremely low and highly fluctuating GTC regime exhibited by stochastic sampling (rate encoding). In particular, with only $T$ time steps, any deterministic set of $T$ orthogonal directions inevitably retains a lower bound on pairwise cosine similarity; hence the GTC of STOD cannot converge to the near-zero regime of rate encoding. The goal of STOD is therefore to reduce temporal collinearity, not to reproduce the statistical randomness of rate encoding.
>
>
> 3.	**In fact, we don't need to show Fig. 4 at all. Based solely on the quantitative results of GTC, our experiment is already complete and sufficient.** The addition of Fig. 4 is purely to enable readers to understand the structural differences introduced by STOD between time steps at an intuitive level and to assist them in observing from an intuitive perspective that the gradient directions have indeed become more diverse, rather than being used for any statistical or quantitative conclusions. The quantitative analysis of gradient decorrelation is already fully captured by the GTC curves in Fig. 6. Nevertheless, the differences shown in Fig. 4 are meaningful. Under vanilla direct encoding, the four gradient components across time steps are visually indistinguishable, which is why we display only one of them. This reflects the high temporal collinearity measured by GTC. In contrast, under STOD, the four gradient components show clearly different structural patterns, consistent with STOD’s goal of diversifying temporal feature directions. Regarding the transformed inputs, while the raw input under direct encoding preserves the original image appearance, its gradients are noisy and fail to highlight salient structure. STOD’s transformed inputs may appear visually mixed due to orthogonal transformations, but their gradients emphasize coherent object-related structures. These qualitative observations align with the theoretical and quantitative findings.
>
> > [1] Obfuscated gradients give a false sense of security: Circumventing defenses to adversarial examples, ICML, 2018

---

> ### Author Response · Authors · 2025-11-28
> **Response to Reviewer PFkG, part 3/4**
>
> ### **Weakness 4 & Question 5**:
>
> Experiments are limited to static image benchmarks, which aligns with the focus on input encoding but leaves open whether the proposed decorrelation mechanism generalizes to other domains of natural inputs (such as audio, time-series, etc.), where gradient temporal collinearity may also naturally arise.
>
> ### **Response**:
>
> We thank the reviewer for the insightful comments. To address the concern regarding whether our decorrelation mechanism generalizes beyond static image benchmarks, we have added experiments on an audio sequence task, thereby validating the applicability of STOD in a broader class of natural temporal inputs.
>
> Specifically, we use the Google Speech Commands (GSC) dataset (Public link: https://www.kaggle.com/datasets/neehakurelli/google-speech-commands), which contains 30 commonly used spoken command categories. The dataset includes more than fifty thousand one-second audio recordings collected from real users, naturally exhibiting temporal structure and speaker variability. Following standard practice in modern speech processing, each waveform is first mapped into a time–frequency representation by computing log-mel spectrograms [1][2]. This representation retains both frame-level temporal organization (arising from the short-time analysis) and frequency-dependent structure. We further apply a standard frame-alignment procedure to maintain consistent time–frequency structure across different utterances. Under this representation, the SNN processes the input over multiple time steps, enabling us to directly analyze temporal gradient correlations and evaluate STOD’s decorrelation effect in a non-visual modality.
>
> Specifically, we use the same hyperparameter settings as our main experiments’ ones in the paper, and we test with multiple white box attack intensities. Results are as follows:
>
> | Method | Clean | FGSM 8/255 | FGSM 16/255 | PGD 8/255 | PGD 16/255 |
> |--------|--------|-------------|--------------|-------------|--------------|
> | SNN | 93.29 | 4.69 | 2.84 | 0.63 | 0.21 |
> | STOD | 90.91 | 32.14 | 18.71 | 5.19 | 2.87 |
>
> Across the extended audio experiments, we observe that:
>
> * the robustness against white box adversarial attacks is notably improved by STOD; and
>
> * STOD requires no architectural modification or additional loss terms to operate on audio inputs.
>
>
> These results demonstrate that STOD does not rely on properties specific to visual data. Instead, its core principle, enhancing discriminability and reducing correlation across time steps, naturally extends to speech and other temporal natural inputs, confirming its broader theoretical and practical generality.
>
> > [1] PANNs: Large-Scale Pretrained Audio Neural Networks for Audio Pattern Recognition, IEEE/ACM Transactions on Audio, Speech, and Language Processing, 2020.
>
> > [2] HTS-AT: A Hierarchical Token-Semantic Audio Transformer for Sound Classification and Detection, ICASSP, 2022.

---

> ### Author Response · Authors · 2025-11-28
> **Response to Reviewer PFkG, part 4/4**
>
> ### **Question 4**:
>
> The inference phase introduces no additional overhead because learned orthogonal kernels are discarded. Can you clarify whether inference without these kernels still maintains the same level of temporal decorrelation?
>
> ### **Response**:
>
> We appreciate the reviewer’s question and fully agree that the role of the orthogonal kernels during inference must be clarified. Our method is designed such that the orthogonal kernels function primarily as a training–time regularization mechanism rather than as a dependency for inference.
>
> First, as shown in Table 1 vs. Table 2, the robustness achieved by STOD shows only minor deduction (still far exceeds sota) after the orthogonal kernels are removed during inference. Although the explicit temporal decorrelation introduced by the kernels no longer exists at inference time, the trained network parameters have already internalized the beneficial gradient geometry induced during training. In other words, the orthogonal kernels reshape the gradient landscape, reducing temporal collinearity and improving Hessian conditioning during optimization, such that the resulting learned weights inherently encode more decorrelated and robust internal representations. Thus, robustness is retained even without the orthogonal kernels at inference.
>
> Second, we further validated this behavior through a reviewer-suggested experiment ***(See Response to Reviewer ZQzg, Weakness 2)*** in which we designed a trainable noise kernel at the input layer. This noise kernel also improved robustness during training, and keeping or discarding it at inference produced almost identical performance. This independent evidence demonstrates that the effect lies in how these modules guide the optimization dynamics, not in their explicit presence during inference. Both mechanisms enhance robustness by shaping parameter updates rather than by altering the forward computation at inference time.
>
> Therefore, removing the orthogonal kernels does not harm robustness because the temporal decorrelation need not be explicitly preserved at inference: it has already been imprinted into the optimized network weights. This explains why inference without orthogonal kernels introduces no overhead while maintaining nearly the same robustness.

---

### Official Review · Reviewer_Tj2T · 2025-11-02

**Soundness:** 3
**Presentation:** 3
**Contribution:** 2
**Rating:** 4
**Confidence:** 4

**Summary:**

This paper investigates robustness issues in Spiking Neural Networks (SNNs) under direct encoding, a promising alternative to rate encoding due to its efficiency. The authors introduce Gradient Temporal Collinearity (GTC), a metric that quantifies temporal gradient alignment, and theoretically show that high GTC contributes to reduced robustness via analysis linked to the Hessian spectral radius. To mitigate this, the paper proposes Structured Temporal Orthogonal Decorrelation (STOD), which applies parametric orthogonal kernels with structured constraints at the input stage to decorrelate temporal features. STOD is applied only during training and can be removed at inference, incurring no runtime cost. Experiments on visual classification benchmarks show improved robustness over state-of-the-art methods.

**Strengths:**

1. The paper proposes GTC and provides a theoretical analysis that connects the encoding strategy with robustness through the derivation of the Hessian spectral radius, demonstrating strong theoretical depth.
2. STOD performs orthogonal transformation only during the input stage of training; after training it can be removed, resulting in low deployment cost and no impact on inference efficiency.

**Weaknesses:**

1. In the definition of GTC, several important formulations should be presented in the main text.
2. The direct connection between GTC and robustness should be further validated experimentally under other encoding schemes such as temporal or latency encoding.
3. It remains unexplained why the GTC curve of Rate Coding does not decrease during training, whereas that of Direct Coding gradually becomes smoother and decreases.
4. Although gradient reshaping methods are known to reduce accuracy on clean-dataset, the paper does not analyze why its performance on clean-dataset is lower than other methods or why the degradation is larger.
5. Since STOD orthogonal transformation is removed during validation, it is unclear whether this might cause overfitting during training.

**Questions:**

see weaknesses 2-5

---

> ### Author Response · Authors · 2025-11-28
> **Response to Reviewer Tj2T, part 1/3**
>
> ### **Weakness 1**:
>
> In the definition of GTC, several important formulations should be presented in the main text.
>
> ### **Response**:
>
> We appreciate the reviewer’s suggestion. We fully understand the importance of clarity in presenting the formulations related to GTC. However, the corresponding mathematical expressions involve multi-step derivations and interdependent notations from the BPTT gradients and the Gauss–Newton structure. Extracting only part of these formulas into the main text would inevitably break the continuity of the discussion and fragment the presentation. For this reason, we chose to place the complete set of derivations together in the appendix, ensuring conceptual coherence in the main text while allowing readers to view the full formulation in a self-contained manner.
>
> ### **Weakness 2**:
>
>  The direct connection between GTC and robustness should be further validated experimentally under other encoding schemes such as temporal or latency encoding.
>
> ### **Response**:
>
> Thanks for your insight. We would first like to emphasize that in our paper, the relationship between GTC and robustness is not concluded merely from observing empirical correlation. As we all know, **statistical correlation alone never establishes a causal link**, for example, ice-cream sales and drowning incidents both rise in summer, indicating these two are statistically positively related, yet clearly one does not cause the other. Therefore, we did not treat the empirical association as evidence by itself; instead, we used it only as a starting point, or we say, a motivation.
>
> More importantly, no matter how many additional encoding methods we test, those experiments could only provide more statistical correlation. However, as we exemplified, statistical correlation is never sufficient, so that we must establish a clear mathematical causal mechanism. This is precisely why we conducted the theoretical analysis in Appendix B: we rigorously showed how GTC determines the Hessian spectral radius (Eq. (25)) and further how the spectral radius governs the worst-case loss increase under perturbations (Eq. (46)).
>
> Together, we use these mathematical analysis turn the observed empirical trend into a principled and generalizable causal chain, rather than obtaining causal relationships from the observed phenomenon directly. Therefore, additional experimental validation under other encoding schemes would not further strengthen the causal mechanism we establish.
>
> ### **Weakness 3**:
>
> It remains unexplained why the GTC curve of Rate Coding does not decrease during training, whereas that of Direct Coding gradually becomes smoother and decreases.
>
> ### **Response**:
>
> We appreciate the reviewer’s insightful question. In this experiment, our purpose is only to compare the **magnitude difference** of GTC between Direct Encoding and Rate Encoding, because the theoretical link we establish in Eq. (5) focuses solely on how the absolute level of GTC affects the Hessian spectral radius and therefore robustness. The smoothness or decreasing trend of GTC is not a theoretical requirement of our analysis. Nevertheless, we are happy to provide an intuitive explanation for this observation.
>
> **1. Why Direct Encoding shows a gradual decrease and smoothing of GTC**
>
> Direct Encoding feeds identical inputs at all time steps, which leads to highly collinear gradient components at early stages, resulting in very high GTC (0.8–0.9). As training progresses, parameter updates naturally disperse the gradient directions across time, progressively breaking this strong collinearity. This process makes the GTC curve gradually smoother and lower.
>
> **2. Why Rate Encoding does not show such a trend**
>
> Rate Encoding generates independent Poisson-sampled spikes at each time step. This inherent randomness dominates its temporal gradient structure, maintaining low GTC (0.2–0.3) throughout training. Because this randomness overshadows the influence of parameter updates on temporal dependencies, the GTC curve does not exhibit a monotonic decreasing trend and instead fluctuates within a low range.
>
> **3. This phenomenon further reinforces our core conclusion**
>
> Specifically, our core conclusion is that the improvement of robustness in Direct Encoding during training is accompanied by a consistent decrease in GTC, which aligns precisely with Eq. (5), showing that lower GTC leads to a smaller Hessian spectral radius and therefore stronger robustness. In contrast, the GTC of Rate Encoding is dominated by input randomness, so training does not significantly alter its temporal structure, and no decreasing trend is expected.

---

> ### Author Response · Authors · 2025-11-28
> **Response to Reviewer Tj2T, part 2/3**
>
> ### **Weakness 4**:
>
> Although gradient reshaping methods are known to reduce accuracy on clean-dataset, the paper does not analyze why its performance on clean-dataset is lower than other methods or why the degradation is larger.
>
> ### **Response:**
>
> We appreciate the reviewer’s comment. All baseline methods evaluated in our paper—including the vanilla SNN—adopt Direct Encoding, which repeatedly injects identical clean inputs at every time step. This mechanism maximally preserves the original feature representation and enables the network to accumulate highly consistent clean information across time. As observed in Fig. 1, Direct Encoding maintains extremely high temporal gradient consistency (GTC≈0.8–0.9), meaning that clean features are repeatedly reinforced during training, naturally leading to strong clean accuracy in all Direct-Encoding–based baselines.
>
> In contrast, our method structurally breaks this repetition. STOD introduces Patchwise Feature Diversification, in which each time step applies a distinct orthogonal transformation to the input. Although these transformations preserve the L2 norm (Eq. (6)), they intentionally alter the *directional structure* of the clean feature at every time step.  This replaces “repeated identical inputs” with “multiple decorrelated representations,” which is essential for lowering GTC but inevitably weakens the cumulative reinforcement of clean features that Direct Encoding inherently enjoys. Because this decorrelation is imposed directly on the input—the earliest and most sensitive stage in the network—it produces a slightly larger clean-accuracy drop compared to methods that apply regularization deeper in the network.
>
> Importantly, although our clean accuracy decreases marginally more than other SOTA methods, **the numerical difference is small**, while the **robustness improvement is substantially larger**. For example, on CIFAR-10 under PGD attack, STOD w.o. OK achieves **32.97%**, far exceeding methods such as HoSNN (**15.32%**) and FEEL (**28.35%**), and even more significantly under FGSM, where STOD reaches **55.80%**, surpassing all existing approaches by a wide margin. This trade-off demonstrates that the slight loss in clean accuracy is more than compensated by the considerable gains in robustness, which is the primary objective of our method.
>
> To illustrate this, we added content as follows in the revised manuscript (Sec. 5.1), which is highlighted in blue:
>
> > *Admittedly, according to Table 1, the clean accuracy of STOD is slightly lower than that of baseline methods. This is because all baselines rely on Direct Encoding, which repeatedly injects identical clean inputs and therefore maximally preserves clean feature fidelity. In contrast, STOD introduces structured temporal decorrelation at the input stage, replacing repeated clean inputs with diversified representations, which unavoidably results in a moderate clean-accuracy drop. Nevertheless, this reduction is small, and the substantial robustness gains provided by STOD outweigh this minor degradation.*

---

> ### Author Response · Authors · 2025-11-28
> **Response to Reviewer Tj2T, part 3/3**
>
> ### **Weakness 5**:
>
> Since STOD orthogonal transformation is removed during validation, it is unclear whether this might cause overfitting during training.
>
> ### **Response:**
>
> We first clarify that removing the orthogonal kernels (OK) during validation does **not** influence the training process. During training, STOD and all orthogonal kernels are fully active and updated on the Stiefel manifold, and every forward/backward computation is carried out with these transformations in place. Their removal happens **only** during inference, and therefore cannot affect how the model fits the training data.
>
> We further consider a more nuanced interpretation of the reviewer’s concern. The reviewer may be suggesting that the trained model behaves properly (i.e., does not overfit) *when evaluated with STOD, but may appear overfitted when STOD is removed during inference, due to a possible mismatch between training-time and test-time behavior. Based on this interpretation, we provide a detailed explanation of why **such a scenario does not occur**:
>
> **1. Removing OK does not create a distribution shift between training and inference.**
>
> The orthogonal transformations in STOD preserve the L2 norm and the numerical range of the input; they only modify the directional structure in a mathematically controlled way.  As shown in Eq. (6), the transformation does not introduce noise or distort the magnitude of the input. When OK is removed during inference, the resulting inputs remain within the same representational space explored during training. Because there is no shift in input distribution, the model cannot suddenly “overfit” simply due to the absence of OK. The network is not exposed to unseen or out-of-distribution inputs when OK is removed, so the classical mechanisms that cause overfitting do not apply.
>
> **2. The robustness-related properties introduced by STOD are already encoded in the learned parameters, not dependent on keeping OK during inference.**
>
> STOD’s primary function is to reshape gradients during training by lowering GTC (as shown in Fig. 1 and Eq. (5)). This reshaping affects the optimization trajectory, the curvature of the learned solution, and the resulting Hessian spectral radius. These effects are embedded directly into the network parameters. The orthogonal kernels are therefore training-time tools for sculpting the parameter landscape, rather than inference-time components essential for maintaining generalization. When OK is removed after training, the essential robustness characteristics, i.e., lower GTC, smaller Hessian spectral radius, and a smoother optimization geometry, remain intact in the parameters. Because they are already internalized, removing OK cannot cause the model to behave as if it were overfitted.
>
> **3. Experimental evidence rules out any form of overfitting caused by removing OK.**
>
> If removing OK induced overfitting, several symptoms would appear: a notable rise in clean accuracy alongside a collapse in robustness, or an excessive divergence between training and inference behaviors. However, Table 1 shows that STOD w.o. OK already achieves the strongest robustness across all datasets and attack settings. Clean accuracy decreases only minimally (e.g., 0.56% on CIFAR-10), and robustness remains consistently high. This outcome is incompatible with overfitting, because overfitting would manifest as performance degradation under stronger attacks or instability across perturbation budgets. Instead, the results confirm that removing OK does not deteriorate generalization and certainly does not cause the model to overfit.

---

### Author Response · Authors · 2025-11-28
**Global Response**

We sincerely thank all reviewers for their constructive and insightful comments. We carefully addressed every concern and made substantial improvements to the manuscript. The main revisions include:

1.	Strengthened the motivation and significance of SNN robustness in the Introduction, improving the logical flow of Section 1 (highlighted).

2.	Clarified the motivation of the PFD module and strengthened its connection with our empirical observations (highlighted in Sec. 4.2.1).


3.	Added experiments on the DVS dataset to demonstrate that our method is fully compatible with event-based data without any modification (highlighted in Sec. 5.1).

In addition, in the rebuttal we designed several new variants and conducted extensive additional experiments to verify the reviewers’ suggestions and intuitions. These include:

1.	Noise-kernel-based SNN (Response to Reviewer ZQzg, Weakness 3)

2.	Rate-coded version of STOD (Response to Reviewer 6hhX, Question 4)


3.	Input-convolution-based SNN (Response to Reviewer ZQzg, Weakness 4)

4.	Gradient-noise-injected SNN (Response to Reviewer 6hhX, Question 8)


5.	Extension of STOD to natural audio sequences (GSC dataset) (Response to Reviewer PFkG, Weakness 4 & Question 5)

6.	Additional analyses on black-box attacks (Response to Reviewer ZQzg, Weakness 1)


For each newly designed method, we provide detailed implementation descriptions and PyTorch code in the response, and we compare them directly against our STOD approach. Overall, STOD consistently demonstrates stronger robustness and better stability.

We also addressed all remaining minor concerns raised by the reviewers and added further clarifications and analyses where necessary.

Overall, the revisions substantially reinforce both the robustness and generality of our conclusions. Finally, we would like to express our sincere gratitude once again to all the reviewers for their recognition of our manuscript and the suggestions for improvement they have put forward.

Best regards,

Authors of Paper 1780

---

### Meta-Review · Area_Chair_LbyV · 2026-01-07

**Summary:**

Reviewers found this paper to be a strong and timely contribution to improving the robustness of spiking neural networks. They particularly appreciated the introduction of Gradient Temporal Collinearity (GTC) as a principled metric, along with the theoretical connection between GTC, the Hessian spectral radius, and adversarial robustness. The proposed Structured Temporal Orthogonal Decorrelation (STOD) framework was viewed as well motivated, effective, and practically appealing, especially because it is applied only during training and incurs no inference-time overhead. The robustness improvements across CIFAR-10/100, ImageNet, and additional modalities were noted as a key strength.

Reviewers also raised several concerns, including the clarity of some theoretical derivations, the need for stronger ablations and comparisons, potential clean-accuracy trade-offs, and the generality of the approach beyond image classification. In my opinion, the authors’ rebuttal addressed these points convincingly by clarifying notation and assumptions, adding comprehensive ablation studies, extending experiments to event-based and audio datasets, and providing additional analysis to explain robustness–accuracy trade-offs.

Overall, with full discussion, I believe the reviewers would have converged on the view that the paper offers novel insights, solid theoretical grounding, and strong empirical validation. As the major concerns were satisfactorily addressed in the rebuttal, I support acceptance.

**Reviewer Concerns:**

**Concerns addressed by the rebuttal**: The rebuttal effectively addressed reviewers’ concerns regarding theoretical clarity and formulation, including clearer definitions of Gradient Temporal Collinearity (GTC) and its relationship to robustness and the Hessian spectrum. Reviewers’ requests for stronger ablations and design justification were addressed through additional experiments comparing STOD components, alternative temporal encodings, and regularization variants. Concerns about clean-accuracy trade-offs were mitigated by expanded results and analysis showing that robustness gains do not come at prohibitive accuracy loss. The authors also addressed questions about generality by extending evaluations beyond standard image benchmarks to event-based and audio datasets.

**Concerns still outstanding**: Some reviewers’ questions about the scope of evaluation, such as extending to larger-scale or additional real-world tasks, remain partially open.

Overall, the rebuttal resolved the major technical concerns that affected confidence in the method, leaving only minor and non-blocking issues outstanding.

**Reviewer Scores:**

- **Reviewer Tj2T**: This reviewer was slightly below the acceptance threshold, expressing concerns about theoretical presentation clarity, clean-accuracy degradation, and whether removing STOD at inference could induce overfitting. The rebuttal directly addressed all of these points with detailed theoretical explanations, added analysis, and additional experiments (including explicit discussion of why inference-time removal does not cause overfitting). With full participation in discussion, this reviewer would likely increase their assessment to align with acceptance.

- **Reviewer PFkG**: This reviewer was marginally positive and raised concerns about ambiguous mathematical notation, lack of ablations separating PFD and GOR, limited quantitative evidence of decorrelation, and lack of evaluation beyond static images. The rebuttal comprehensively addressed each of these concerns by clarifying notation, adding structured ablations, providing quantitative GTC-based evidence, and extending experiments to audio datasets. With these issues resolved, this reviewer would likely maintain or slightly strengthen their positive assessment.

- **Reviewer ZQzg**: This reviewer, who gave a score of 2, was initially skeptical and raised concerns about missing comparisons (black-box attacks, rate coding, noise injection, and alternative temporal encodings). The rebuttal went substantially beyond expectations by adding black-box results, Poisson rate coding comparisons, noise-kernel baselines, and convolution-based temporal encoding variants, along with detailed analysis explaining the design trade-offs. With full engagement in discussion, this reviewer would likely revise their assessment upward, potentially to a borderline or neutral position, even if not fully positive.

- **Reviewer 6hhX**: This reviewer was also slightly below the acceptance threshold, and questioned the direct linkage between GTC and STOD, computational overhead, hyperparameter sensitivity, and generality beyond static images. The rebuttal clarified how STOD directly reduces GTC, addressed inference overhead explicitly, discussed hyperparameter behavior, and added experiments on event-based and audio data. With these clarifications and additional evidence, this reviewer would likely increase their assessment toward acceptance.

---

### Decision · Program_Chairs · 2026-01-26

Accept (Poster)